# Phosphorylation of iRhom2 at the plasma membrane controls mammalian TACE-dependent inflammatory and growth factor signalling

Adam Graham Grieve[†], Hongmei Xu[†‡], Ulrike Künzel, Paul Bambrough, Boris Sieber, Matthew Freeman*

Sir William Dunn School of Pathology, University of Oxford, Oxford, United Kingdom

**Abstract** Proteolytic cleavage and release from the cell surface of membrane-tethered ligands is an important mechanism of regulating intercellular signalling. TACE is a major shedding protease, responsible for the liberation of the inflammatory cytokine TNFα and ligands of the epidermal growth factor receptor. iRhoms, catalytically inactive members of the rhomboid-like superfamily, have been shown to control the ER-to-Golgi transport and maturation of TACE. Here, we reveal that iRhom2 remains associated with TACE throughout the secretory pathway, and is stabilised at the cell surface by this interaction. At the plasma membrane, ERK1/2-mediated phosphorylation and 14-3-3 protein binding of the cytoplasmic amino-terminus of iRhom2 alter its interaction with mature TACE, thereby licensing its proteolytic activity. We show that this molecular mechanism is responsible for triggering inflammatory responses in primary mouse macrophages. Overall, iRhom2 binds to TACE throughout its lifecycle, implying that iRhom2 is a primary regulator of stimulated cytokine and growth factor signalling.

*For correspondence: matthew. freeman@path.ox.ac.uk

[†]These authors contributed equally to this work

Present address: [‡]National Key Laboratory of Medical Immunology, Institute of Immunology, Second Military Medical University, Shanghai, China

## Introduction

Signalling ligands are often synthesised as transmembrane domain (TMD) containing precursors. Upon their delivery to the plasma membrane, protease activity is required to shed the bioactive extracellular domain to allow signal release and subsequent binding to receptors on signal-receiving cells. TACE (also known as ADAM17) is a primary shedding enzyme and as such regulates multiple signalling pathways through its ability to cleave and release numerous membrane-tethered signalling ligands and receptors (*Peschon et al., 1998*). Of particular interest, it controls the shedding of TNFα, the principal inflammatory cytokine (*Black et al., 1997*; *Moss et al., 1997*), amphiregulin, TGFα and most other ligands of the epidermal growth factor (EGF) family (*Sunnarborg et al., 2002*; *Sahin et al., 2004*). Disturbances in these cytokine and growth factor signalling pathways are hallmarks of inflammation and cancer, respectively, as well as other diseases. This illustrates the potentially dangerous consequence of unregulated TACE activity and explains the tight post-translational regulation to which TACE is subject.

TACE is first synthesised in the endoplasmic reticulum (ER) as an immature form containing an inhibitory pro-domain that prevents its proteolytic activity. We and others identified the iRhom proteins, catalytically inactive members of the rhomboid-like superfamily, as essential regulators of TACE maturation (*Adrain et al., 2012*; *McIlwain et al., 2012*; *Siggs et al., 2012*; *Issuree et al., 2013*). We reported that iRhoms control transport of TACE from the ER to the Golgi apparatus, where removal of its pro-domain by pro-protein convertases such as furin occurs (*Endres et al.,*

**eLife digest** Injury or infection can cause tissues in the body to become inflamed. The immune system triggers this inflammation to help repair the injury or fight the infection. A signal molecule known as TNF – which is produced by immune cells called macrophages – triggers inflammation. This protein is normally attached to the surface of the macrophage, and it only activates inflammation once it has been cut free.

An enzyme called TACE cuts and releases TNF from the surface of macrophages. This enzyme is made inside the cell and is then transported to the surface. On the way, TACE matures from an inactive form to a fully functional enzyme. Previous work revealed that a protein called iRhom2 controls TACE maturation, but it has been unclear whether iRhom2 affects TACE in any additional ways.

Grieve et al. studied the relationship between iRhom2 and TACE in more detail. The experiments show two new roles for iRhom2: in protecting TACE from being destroyed at the cell surface, and prompting TACE to release TNF to trigger inflammation. Injury or infection causes small molecules called phosphate groups to be attached to iRhom2 in macrophages, which causes TACE to release TNF.

The findings of Grieve et al. provide the first evidence that iRhom2 influences the activity of TACE throughout the enzyme's lifetime. Excessive inflammation, often triggered by the uncontrolled release of TNF, can lead to rheumatoid arthritis, cancer and many other diseases. Therefore, iRhom2 could be a promising new target for anti-inflammatory drugs that may help to treat these conditions.

2003). In this way, iRhoms regulate the conversion of TACE from an inactive immature form to a mature proteolytically competent shedding enzyme (*Adrain and Freeman, 2012*; *Freeman, 2014*; *Lemberg and Adrain, 2016*). Without iRhoms there is no TACE maturation and therefore no TACE activity (*Christova et al., 2013*; *Li et al., 2015*). Of the two mammalian iRhoms, iRhom1 is broadly expressed, whereas macrophages express only iRhom2. Since macrophages are the major TNFα-releasing cell type (*Parameswaran and Patial, 2010*), this makes iRhom2 an important regulator of inflammation. Accordingly iRhom2 knock-out mice have profound inflammatory defects, are sensitive to bacterial infection and are resistant to LPS-induced toxic shock and the development of inflammatory arthritis (*Adrain et al., 2012*; *McIlwain et al., 2012*; *Issuree et al., 2013*). More recently iRhom2 was shown to modulate innate immunity to DNA viruses by regulating the ER-to-Golgi transport and the stability of the immune adaptor STING (*Luo et al., 2016*). Although there are also hints that iRhom2 might regulate TACE protein stability (*Maney et al., 2015*), this has not been explored in detail.

iRhoms comprise a seven transmembrane rhomboid-like domain, with a short luminal C-terminus, a well conserved loop between TMDs 1 and 2 (termed the iRhom homology domain, IRHD), and a long cytoplasmic N-terminus (*Lemberg and Freeman, 2007*, *2016*). There is accumulating physiological and cell biological evidence that mutation or deletion of the cytoplasmic N-terminus of iRhoms leads to alterations in TACE function (*Johnson et al., 2003*; *Blaydon et al., 2012*; *Saarinen et al., 2012*; *Brooke et al., 2014*; *Hosur et al., 2014*; *Siggs et al., 2014*; *Liu et al., 2015*; *Maney et al., 2015*). Moreover, iRhoms act downstream of G-protein coupled receptor (GPCR) signalling in the transactivation of EGFR signalling (*Christova et al., 2013*; *Li et al., 2015*). Taken together, these observations indicate that the cytosolic N-terminal domain of iRhoms regulates TACE, but the underlying mechanisms are unclear. Importantly, TACE-induced ligand shedding is rapid: it is activated within ten minutes by physiological and pharmacological stimulation (*Le Gall et al., 2010*), and this is compatible with the timescale on which iRhoms have been reported to influence TACE substrate selection (*Maretzky et al., 2013*). However, this speed of TACE activation is not easily reconciled with its relatively slow ER-to-Golgi trafficking and maturation (3–6 hr, [*Schlöndorff et al., 2000*]), the process by which iRhoms are known to control TACE.

Here we resolve this apparent paradox by demonstrating that iRhom2-mediated TACE maturation represents just the first step in the relationship between them. Once mature TACE has been

trafficked to the cell surface, it requires a further iRhom2-dependent activation step, mediated by phosphorylation of the iRhom2 cytoplasmic domain. We reveal two unanticipated molecular roles for iRhom2 beyond its described function in TACE maturation. First, iRhom2 interacts with mature TACE at the plasma membrane, simultaneously stabilising TACE from lysosomal degradation and limiting its proteolytic activity. Second, iRhom2 acts as a signalling hub whereby phosphorylation of its N-terminus and subsequent 14-3-3 protein binding controls stimulated TACE proteolytic activity and ligand shedding. We show that this regulation of mature TACE activity occurs not only in immortalised cell models but also regulates inflammatory TNFα release from primary macrophages. Overall, we conclude that iRhom2 binds to TACE and controls its function at multiple stages, indicating that iRhom2 can be treated as a multifunctional regulatory subunit of TACE.

## Results

### TACE activity, but not TACE maturation, is regulated by iRhom2 phosphorylation

iRhom2 is an essential regulator of TACE maturation and therefore TACE-dependent inflammatory and growth factor signalling (*Adrain et al., 2012*; *McIlwain et al., 2012*; *Siggs et al., 2012*; *Christova et al., 2013*; *Issuree et al., 2013*; *Li et al., 2015*). Accumulating data about the significance of the iRhom2 N-terminus led us to hypothesise that iRhom2 control over TACE was subject to further regulation. We discovered that treatment with phorbol-12-myristate-13-acetate (PMA), a common activator of TACE-dependent signalling, leads to serine phosphorylation of mouse iRhom2 within 15 min (*Figure 1a*). Furthermore, even without stimulation, mass spectrometric analysis revealed multiple phosphorylated serines within the human iRhom2 cytoplasmic N-terminus (*Figure 1b*; *Figure 1—figure supplement 1 a-b*. Combined, this indicates that the iRhom2 N-terminus is subject to a basal level of phosphorylation, which is enhanced by a stimulator of TACE enzyme activity.

We focused on the conserved phosphorylation sites in the N-terminus of iRhom2, combined with phosphosites reported in other screens (http://www.phosphosite.org/). An extensive mutagenesis of mouse iRhom2 was performed, mutating all potential phosphorylation sites in the N-terminus to alanine (referred to as iRhom2$^{pDEAD}$, *Figure 1c*). We found that blocking iRhom2 N-terminus phosphorylation had no effect on iRhom regulation of the ER-to-Golgi transport and subsequent maturation of TACE. Expression of iRhom2$^{pDEAD}$ in iRhom1/2 double knock-out (DKO) MEFs rescued TACE maturation to the same extent as iRhom2$^{WT}$ (*Figure 1d*, grey arrowhead). Therefore iRhom2 phosphorylation is not essential for the maturation of TACE, suggesting that any phosphorylation-dependent function is restricted to an unknown post-TACE maturation role of iRhom2. Notably, a form of iRhom2 C-terminally tagged with a KDEL motif (*Munro and Pelham, 1987*), which enhances ER retrieval from the cis-Golgi, did not support TACE maturation (*Figure 1d*, *Figure 1—figure supplement 2 a-c*, implying that iRhom2 itself must be trafficked beyond the ER to promote TACE maturation. In support of the conclusion that iRhom2 phosphorylation does not control ER-to-Golgi trafficking of TACE, iRhom2$^{WT}$ and iRhom2$^{pDEAD}$, but not iRhom2$^{KDEL}$, were both detected in the Golgi apparatus, co-localising with the cis-Golgi marker GM130 (*Figure 1e*). Finally, consistent with a previous report (*Maney et al., 2015*), we found that iRhom2 can reach the cell surface: using cell surface biotinylation we detected both iRhom2$^{WT}$ and iRhom2$^{pDEAD}$ at the plasma membrane (*Figure 1f*). Overall, this demonstrates that iRhom2 is not limited to the early secretory pathway, but also reaches the plasma membrane, and that the ER-to-Golgi traffic of iRhom2 and TACE is independent of iRhom2 N-terminal phosphorylation.

The independence of TACE maturation from iRhom2 phosphorylation – combined with the plasma membrane localisation of iRhom2 – hinted that phosphorylation might regulate an as yet unknown post-Golgi iRhom2 function. We therefore tested whether iRhom2 phosphorylation was required for TACE shedding activity. We found that iRhom2$^{pDEAD}$ was greatly impaired in its ability to promote TACE-dependent shedding of EGF-like ligands (*Figure 1g–h*) and TNFα (*Figure 1i*) in response to PMA activation. The requirement for iRhom2 phosphorylation was specific to TACE substrates since the shedding of EGF or betacellulin, which depends on the related protease ADAM10 (*Sahin et al., 2004*), was unaffected by absence of iRhom2 phosphorylation (*Figure 1j*). We conclude that phosphorylation of the iRhom2 N-terminus does not regulate its role of controlling ER-to-Golgi

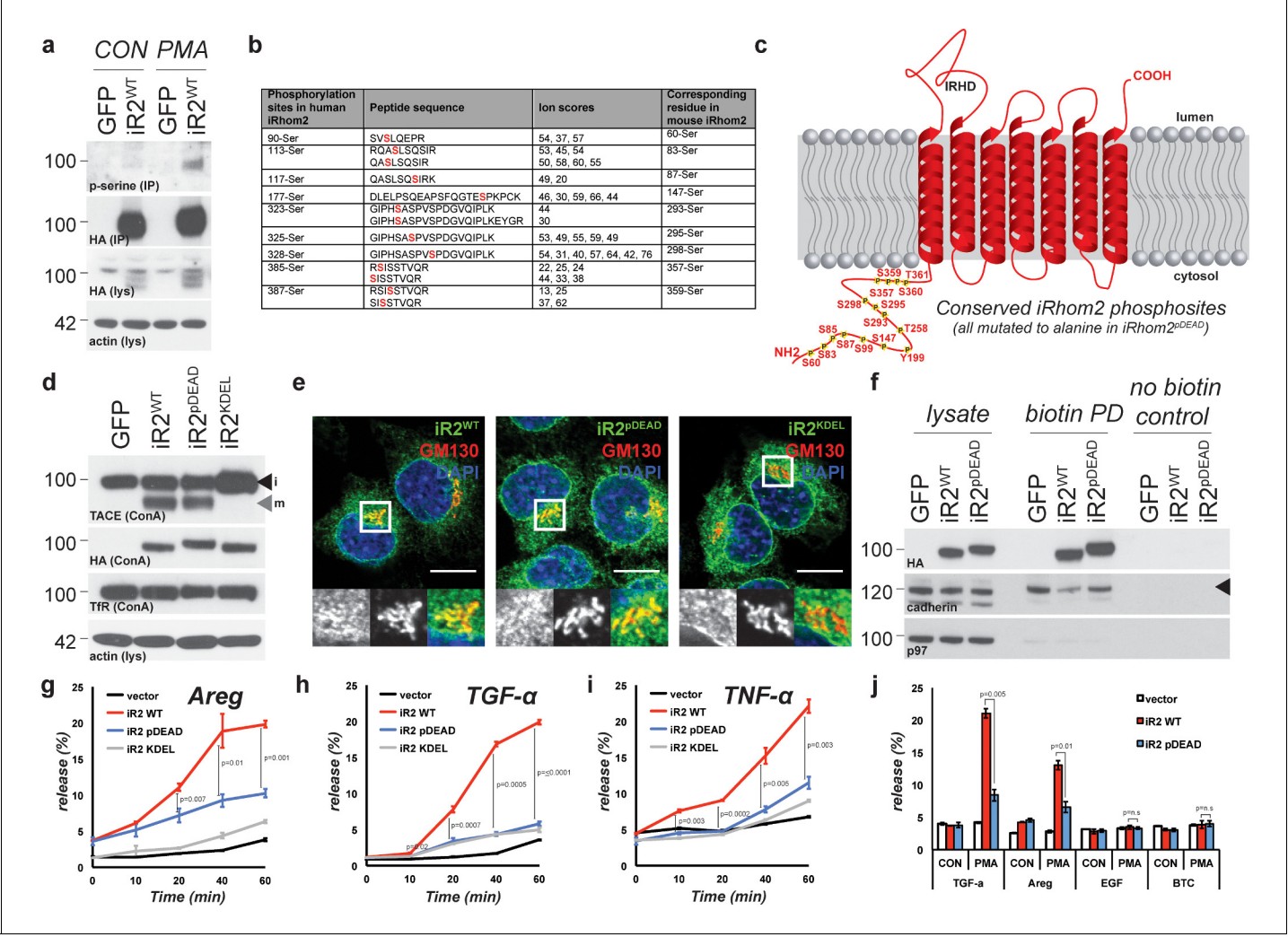

**Figure 1.** TACE activity, but not TACE maturation is regulated by iRhom2 phosphorylation. (**a**) HA immunoprecipitates (IP) and lysates (lys) from HEK293 cells stably expressing GFP or mouse iRhom2[WT]-3xHA were immunoblotted for phosphorylated serine, HA and beta-actin after treatment with or without 200 nM PMA for 15 min. (**b**) Phosphorylation sites identified (in red) in human iRhom2 by iRhom2-3xHA immunoprecipitation and mass spectrometry. Ion scores derive from separate peptides from at least two independent datasets. For comparison, the corresponding conserved residue in mouse iRhom2 is stated (**c**) A model of mouse iRhom2 (in red), with the conserved N-terminal serine/threonine phosphorylation sites (yellow stars) identified in *Figure 1b* and from public source databases. All of these sites have been mutated to alanine in iRhom2[pDEAD]. The iRhom homology domain (IRHD) is indicated between the first two transmembrane domains. (**d**) Lysates from iRhom1/2 double knockout (DKO) mouse embryonic fibroblasts (MEFs) expressing GFP or different versions of HA-tagged iRhom2 enriched for glycoproteins with Concanavalin A (ConA) were immunoblotted for TACE, HA, and transferrin receptor (TfR) as a labelling control. Lysates were probed with beta-actin antibody. Black arrowhead: immature TACE. Grey arrowhead: mature TACE. (**e**) Immunofluorescence of DKO MEFs transduced with different versions of iRhom2-3xHA, labelled with HA (green), GM130 label for the cis-Golgi (red) and DAPI for DNA (blue). GM130-labelled regions (within white boxes) have been magnified. Scale bar = 10 μm. (**f**) Lysates and neutravidin-enriched preparations from MEFs transduced with GFP, or HA-tagged iRhom2[WT] and iRhom2[pDEAD] labelled with (biotin pull down) or without biotin (no biotin control), probed with HA, pan-cadherin (cell surface positive control; arrowhead indicates specific band) and p97 (cell surface negative control) antibodies. (**g–j**) DKO MEFs transduced with GFP or different versions of iRhom2-3xHA transfected with alkaline-phosphatase tagged amphiregulin (Areg), transforming growth factor (TGFα), tumour necrosis factor (TNFα), epidermal growth factor (EGF) or betacellulin (BTC), stimulated for indicated times with 200 nM PMA. Values represent the level of released alkaline phosphatase/total alkaline phosphatase multiplied by 100. Error bars representing standard deviations and values for two-tailed student t-tests are depicted.

The following figure supplements are available for figure 1:

**Figure supplement 1.** Human iRhom2 phosphorylation can be detected by mass spectrometry.

**Figure supplement 2.** Characterisation of iRhom2[KDEL].

trafficking and maturation of TACE, but instead specifically controls the proteolytic activity of mature TACE at the cell surface.

## Mature TACE activity is dependent upon iRhom2 phosphorylation at three distinct sites, which co-ordinate binding to 14-3-3 family proteins

These data demonstrate that phosphorylation of the cytoplasmic N-terminus of iRhom2 specifically regulates the ability of mature TACE to release growth factors and cytokines from the cell surface. To further investigate this newly identified function we performed individual and combined alanine-scanning of the putative phosphorylation sites in iRhom2 (*Figure 2a*, *Figure 2—figure supplement 1a–c*, and data not shown for non-contributing residues). This revealed three distinct sites that strongly contribute to PMA-induced TACE-dependent shedding: Site1 comprises S58 and S60; Site2 comprises S83, S85, and S87; and Site3 comprises S357, S359, S360 and T361 (*Figure 2a* - highlighted in red text, and *Figure 1c*). Assessment of the three phosphorylation site mutants individually (iRhom2$^{Site1}$, iRhom2$^{Site2}$ and iRhom2$^{Site3}$) and in combination (iRhom2$^{Site1-3}$) reveals that they additively regulate TACE-dependent shedding of both TGFα and TNFα (*Figure 2b and c*). PMA is not a physiological trigger for TACE activity, so to confirm the physiological significance of our observations, we showed that GPCR stimulation with histamine triggers TACE-dependent release of amphiregulin and TGFα in an iRhom2 phosphorylation-dependent manner (*Figure 2d*). Finally, we confirmed that stimulated phosphorylation of iRhom2 is lost upon simultaneous alanine mutation of all three of the identified sites required for TACE activity (*Figure 2e*).

We hypothesised that phosphorylation might affect iRhom2 interactions with cytoplasmic partners. In a proteomic screen for proteins that interact with iRhom2, we identified multiple 14-3-3 family proteins binding to the cytoplasmic domain (*Figure 2f*). 14-3-3 proteins bind phosphorylated residues and have roles in signalling (*Fu et al., 2000*), so we further investigated their interaction with iRhom2. Using both iRhom2$^{pDEAD}$ and iRhom2$^{Site1-3}$, we found that 14-3-3 proteins strongly bind to the N-terminus of iRhom2 upon PMA stimulation, in a phosphorylation-dependent manner (*Figure 2g*). Of the seven 14-3-3 isoforms, we could confirm by western blot that all bound phosphorylated iRhom2 apart from 14-3-3α/β, although binding to 14-3-3ζ was weak (*Figure 2—figure supplement 1d–f*; 14-3-3σ antibody failed, data not shown). Last, we found that the ER-Golgi retention mutant, iRhom2$^{KDEL}$, also recruited 14-3-3 proteins upon stimulation with PMA (*Figure 2—figure supplement 1g*), indicating that iRhom2 is phosphorylated throughout the secretory pathway. Overall, these data show that 14-3-3 proteins bind to the functionally important iRhom2 phosphorylation sites.

## iRhom2 phosphorylation and 14-3-3 binding is dependent on active ERK signalling

Our mass spectrometry screen reproducibly identified ribosomal protein S6 kinase alpha-3 (RSK3) and extracellular signal-regulated kinases 1/2 (ERK1/2, also known as MAPK3/1 respectively) as potential binding partners of iRhom2 (*Figure 3a*). Using PMA stimulated TACE-dependent shedding of amphiregulin as a readout, we examined the role of these kinases. We screened the activity of RSK3 with the well described inhibitor, BI-D1870 (*Sapkota et al., 2007*); ERK1/2 activity was screened with a potent inhibitor of upstream activator kinases MEK1/2, U0126 (*Favata et al., 1998*). As PMA stimulation is a known activator of protein kinase C (*Ryves et al., 1991*), the PKC inhibitors Go6976 and Go6983 were used as positive controls (*Martiny-Baron et al., 1993*; *Gschwendt et al., 1996*). This experiment demonstrated that PMA stimulated amphiregulin release is insensitive to RSK inhibition, but reduced upon MEK1/2 inhibition (*Figure 3b*), in line with a previous study (*Xu et al., 2012*). We also tested more physiological stimulation by the GPCR agonist histamine. In this case, too, stimulated release of amphiregulin by TACE was inhibited by U0126 (*Figure 3c*). Importantly, we showed that both PMA and histamine treatments indeed do cause upregulated ERK1/2 activity, monitored through enhanced ERK1/2 phosphorylation (*Figure 3d–f*). These results are consistent with iRhom2 phosphorylation being downstream of ERK1/2, and this was confirmed by the observation that stimulated ERK1/2 phosphorylation is unaffected in DKO MEFs expressing the iRhom2$^{Site1-3}$ transgene (*Figure 3e–f*). Since TACE itself is phosphorylated downstream of ERK1/2 signalling (*Soond et al., 2005*), it is possible that the observed inhibition of TACE-dependent shedding was independent of iRhom2 phosphorylation. We therefore assessed the effects of U0126

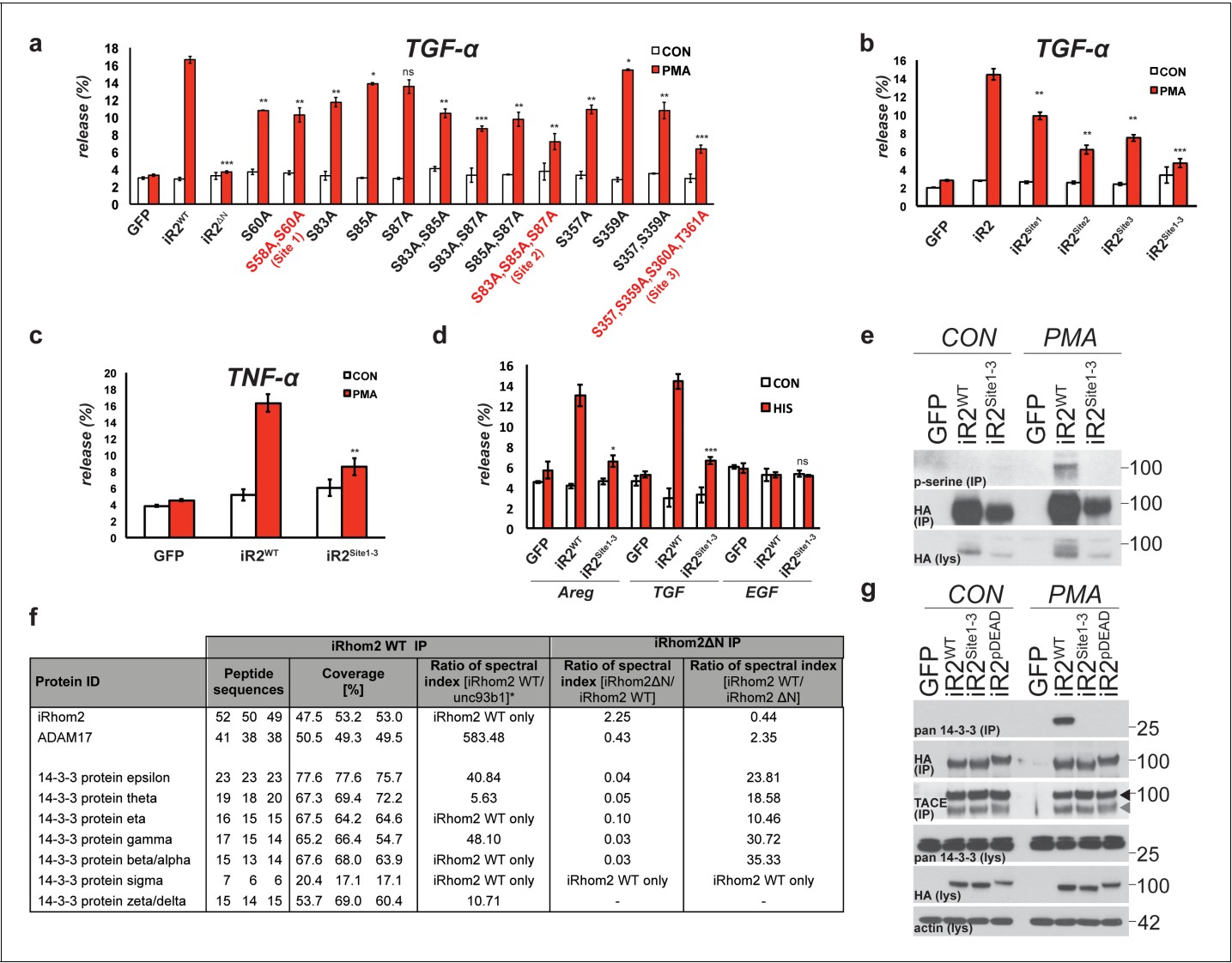

**Figure 2.** Mature TACE activity is dependent upon iRhom2 phosphorylation at three distinct sites, which co-ordinate binding to 14-3-3 family proteins. (a–c) DKO MEFs stably transduced with GFP or different versions of iRhom2-3xHA transfected with alkaline-phosphatase tagged TGFα or TNFα stimulated for 30 min with 200 nM PMA. The contributing residues of Sites 1–3 in iRhom2 are highlighted in red text. (a–c). (d) As stated for (a–c), except DKO MEFs were co-transfected with Areg, TGF or EGF and the histamine HA1 receptor, then stimulated for 60 min with 300 µM histamine a–d. Values represent the level of released alkaline phosphatase/total alkaline phosphatase multiplied by 100. Error bars representing standard deviations and values for two-tailed student t-tests are depicted. ns = $p > 0.05$, *$p < 0.05$, **$p < 0.01$, ***$p < 0.001$. (e) HA immunoprecipitates from HEK293 cells stably expressing GFP or HA-tagged mouse iRhom2$^{WT}$ or iRhom2$^{Site1-3}$ were immunoblotted for phosphorylated serine and HA after treatment with or without 200 nM PMA for 15 min. Lysate was probed with HA antibody. (f) Lysates from HEK293T cells overexpressing HA-tagged human iRhom2$^{WT}$, iRhom2$^{ΔN}$ (lacking aa 1–382) and unc93B1 were subjected to anti-HA immunoprecipitation and analysed by semi-quantitative label-free mass spectrometry. The ratio of the spectral index of iRhom2$^{WT}$ is compared to a control pull-down with the polytopic membrane protein unc93b1 to compensate for unspecific binding to hydrophobic TMDs. The ratio of the spectral index of iRhom2$^{WT}$ and iRhom2$^{ΔN}$ shows the decreased binding of 14-3-3 proteins to iRhom2$^{ΔN}$. (g) HA immunoprecipitations and lysates from DKO MEFs transduced with GFP, HA-tagged mouse iRhom2$^{WT}$ or stated phosphorylation site-to-alanine mutants, treated with or without 200 nM PMA for 30 min, immunoblotted for HA, TACE, pan-14-3-3 and beta-actin.

The following figure supplement is available for figure 2:

**Figure supplement 1.** Analysis of the contribution to TGF shedding of each iRhom2 residue within each phosphorylation site.

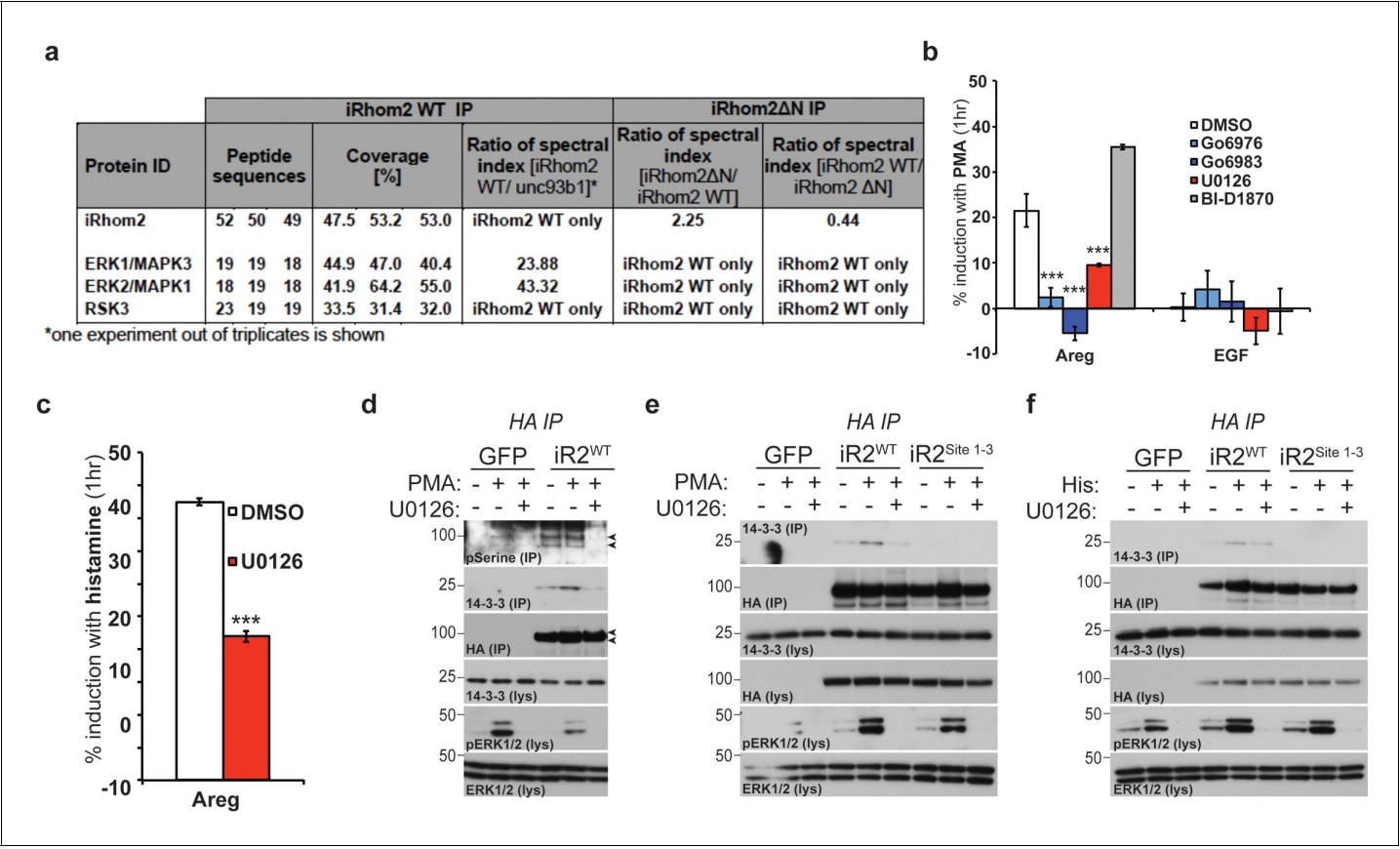

**Figure 3.** iRhom2 phosphorylation and 14-3-3 binding is dependent on active ERK signalling. (a) Lysates from cells overexpressing HA tagged human iRhom2$^{WT}$ and iRhom2$^{\Delta N}$ (lacking aa 1–382) and unc93B1 were subjected to HA immunoprecipitation and analysed by semi-quantitative label-free mass spectrometry. The ratio of the spectral index of iRhom2$^{WT}$ is compared to the control pull-down with unc93b1 to compensate for unspecific binding to hydrophobic TMDs. The ratio of the spectral index of iRhom2$^{WT}$ and iRhom2$^{\Delta N}$ shows the decreased binding of the identified kinases upon deletion of the N-terminus. (b) DKO MEFs stably transduced with HA tagged murine iRhom2$^{WT}$ transfected with alkaline-phosphatase tagged amphiregulin or EGF, pretreated for 1 hr with the indicated kinase inhibitors (20 µM) followed by stimulation for 1 hr with 20 nM PMA. (c) DKO MEFs stably transduced with HA tagged murine iRhom2$^{WT}$ transfected with alkaline-phosphatase tagged amphiregulin and the histamine HA1 receptor were pretreated with U0126 (20 µM) for 1 hr followed by stimulation for 1 hr with 300 µM histamine. In b–c, ***p<0.001. (d-e) HA immunoprecipitates (IP) and lysates (lys) from DKO MEFs transduced with GFP and HA-tagged iRhom2$^{WT}$ (and iRhom2$^{Site1-3}$, in e) pre-treated with U0126 (20 µM) for 1 hr and stimulated with 20 nM PMA for 15 min, immunoblotted for 14-3-3, HA, ERK1/2 and phosphorylated ERK1/2. Arrowheads indicate the iRhom2 phosphoserine positive bands. (f) HA immunoprecipitates (IP) and lysates (lys) from DKO MEFs transduced with GFP and HA-tagged iRhom2$^{WT}$ and iRhom2$^{Site1-3}$, transfected with 4 µg histamine HA1 receptor (per 10 cm plate), pre-treated with U0126 (20 µM) for 1 hr and stimulated with 300 µM histamine for 15 min, immunoblotted for 14-3-3, HA, ERK1/2 and phosphorylated ERK1/2.

directly on serine phosphorylation of iRhom2 and the subsequent recruitment of 14-3-3 proteins. First, we show that phosphorylation of iRhom2 is dependent on MEK1/2 activity (*Figure 3d*). Second, PMA and histamine induced recruitment of 14-3-3 proteins to iRhom2 is abrogated by MEK1/2 inhibition with U0126 (*Figure 3d–f*). We conclude from these data that iRhom2 is phosphorylated downstream of activated ERK signalling, and that the activity of these kinases is required for 14-3-3 recruitment to the N-terminus of iRhom2 and TACE-dependent shedding.

## The amino-terminus of iRhom2 is required for binding to, and stabilisation of, mature TACE

We found that multiple 14-3-3 proteins interact with the iRhom2 N-terminus upon its phosphorylation and that iRhom2 N-terminal phosphorylation strongly promotes TACE activity. The two proteins are known to interact (*Adrain et al., 2012*; *Figure 4—figure supplement 1a*), but the nature of the iRhom2-TACE complex is still largely uncharacterised. Strikingly, iRhom2 lacking its cytoplasmic

domain (iRhom2$^{\Delta NT}$) did not bind mature TACE, although its binding to immature TACE was unaffected (*Figure 4a*). In contrast, deletion of the iRhom homology domain (iRhom2$^{\Delta IRHD}$, *Figure 1c*) strongly impaired immature TACE binding in DKO MEFs (*Figure 4b*). Overall, we could detect no significant difference in the traffic of iRhom2$^{\Delta NT}$ or iRhom2$^{\Delta IRHD}$, relative to their expression level, using cell surface biotinylation (*Figure 4—figure supplement 1b*). These data reveal that the N-terminus of iRhom2 is specifically required for its interaction with mature TACE, but is not required for iRhom2 binding to the immature form of TACE, which instead depends on the iRhom2 IRHD.

Despite its ability to bind immature TACE, iRhom2$^{\Delta NT}$ was unable to support TACE maturation (*Figure 4c*, compare lanes 4 and 7). This is consistent with previous reports (*Siggs et al., 2014*; *Maney et al., 2015*) but establishes an apparent paradox – that iRhom2$^{\Delta NT}$ is able to bind immature TACE but cannot promote its trafficking and maturation. The fact that we did not observe mature TACE could also be explained by degradation of the mature form. To test this, we blocked the two major protein degradation pathways: lysosomal and proteasomal degradation were inhibited with Bafilomycin A1 and MG132, respectively. This showed that iRhom2$^{\Delta NT}$ can in fact support TACE maturation, but in the absence of the iRhom2 cytoplasmic domain, mature TACE is unstable and degraded in lysosomes (*Figure 4c*, compare lanes 7 and 8; *Figure 4—figure supplement 1c*). This result uncovers an unanticipated molecular function for iRhom2, regulating the post-Golgi stability of mature TACE. It also provides an explanation for the requirement for the iRhom2 N-terminus in TACE-dependent stimulated shedding of TGFα (*Figure 2a* and [*Maretzky et al., 2013*]). By comparison, iRhom2$^{\Delta IRHD}$, which is strongly impaired in its ability to bind immature TACE, did not support TACE maturation: the absence of mature TACE was not rescued by either of the degradation inhibitors (*Figure 4c*, compare lanes 4 and 10–12). This implies that, unlike the N-terminus, the iRhom2 IRHD is essential for the ER-to-Golgi traffic and maturation of TACE, further distinguishing the newly discovered post-Golgi function of iRhom2 from its previously reported ER-to-Golgi trafficking role.

Collectively, these results show that the iRhom2 cytoplasmic N-terminus is not needed for ER-to-Golgi trafficking and maturation of TACE, but instead mediates its binding to mature TACE at the plasma membrane. This binding stabilises TACE by preventing its lysosomal degradation.

## Phosphorylation and 14-3-3 protein recruitment uncouples the interaction between iRhom2 and mature TACE to drive its activity at the cell surface

The N-terminus of iRhom2 stabilises mature TACE at the plasma membrane. How does this affect proteolytic activation of TACE? We hypothesised that this might be the step that depends on the N-terminal phosphorylation of iRhom2 and subsequent 14-3-3 protein binding that we had discovered. To test this, we first questioned whether 14-3-3 recruitment to iRhom2 is sufficient to promote TACE activity. The phosphorylation Sites 1 and 2 in iRhom2 were replaced with the R18 peptide sequence that constitutively binds 14-3-3 proteins (*Wang et al., 1999*) (iRhom2$^{R18}$, *Figure 5a*). iRhom2$^{R18}$ not only showed constitutive recruitment of 14-3-3 proteins (*Figure 5b*) but also constitutively elevated TACE activity (*Figure 5c*), indicating that 14-3-3 recruitment to the iRhom2 cytoplasmic domain is indeed sufficient to activate the shedding activity of TACE. Elevated levels of mature TACE did not cause this increase in shedding, as iRhom2$^{R18}$ supported TACE maturation to the same extent as iRhom2$^{WT}$ (*Figure 5d*). Furthermore, the insertion of two R18 peptides in iRhom2 did not affect its plasma membrane levels, assessed by surface biotinylation (*Figure 5—figure supplement 1*). Significantly, we found that this constitutive 14-3-3 binding form binds mature TACE less well than iRhom2$^{WT}$ (*Figure 5e*). Reduction of binding was not caused simply by the loss of the Site 1 and 2 phosphorylation sites in iRhom2$^{R18}$, as iRhom2$^{Site1-2}$ could still bind to mature TACE normally (*Figure 5e*). This suggests that 14-3-3 recruitment to iRhom2 modifies the complex to weaken mature TACE binding.

In the light of the emerging model that 14-3-3 recruitment alters the iRhom2-TACE complex and triggers enzymatic activation of mature TACE, we noted an apparent inconsistency: activation of cells with PMA did not appear to affect binding between iRhom2 and mature TACE (*Figure 2g*). Furthermore, iRhom2$^{WT}$, iRhom2$^{pDEAD}$ and iRhom2$^{Site1-3}$ appeared to bind mature TACE equally (*Figure 2g*). We hypothesised that any alteration in binding might be hard to detect if it were limited to the physiologically active TACE and iRhom complex at the plasma membrane, which represents only a small fraction of total protein. To test this, we investigated the interaction between iRhom2 and TACE specifically at the cell surface. Intact cells on ice were incubated with an antibody that

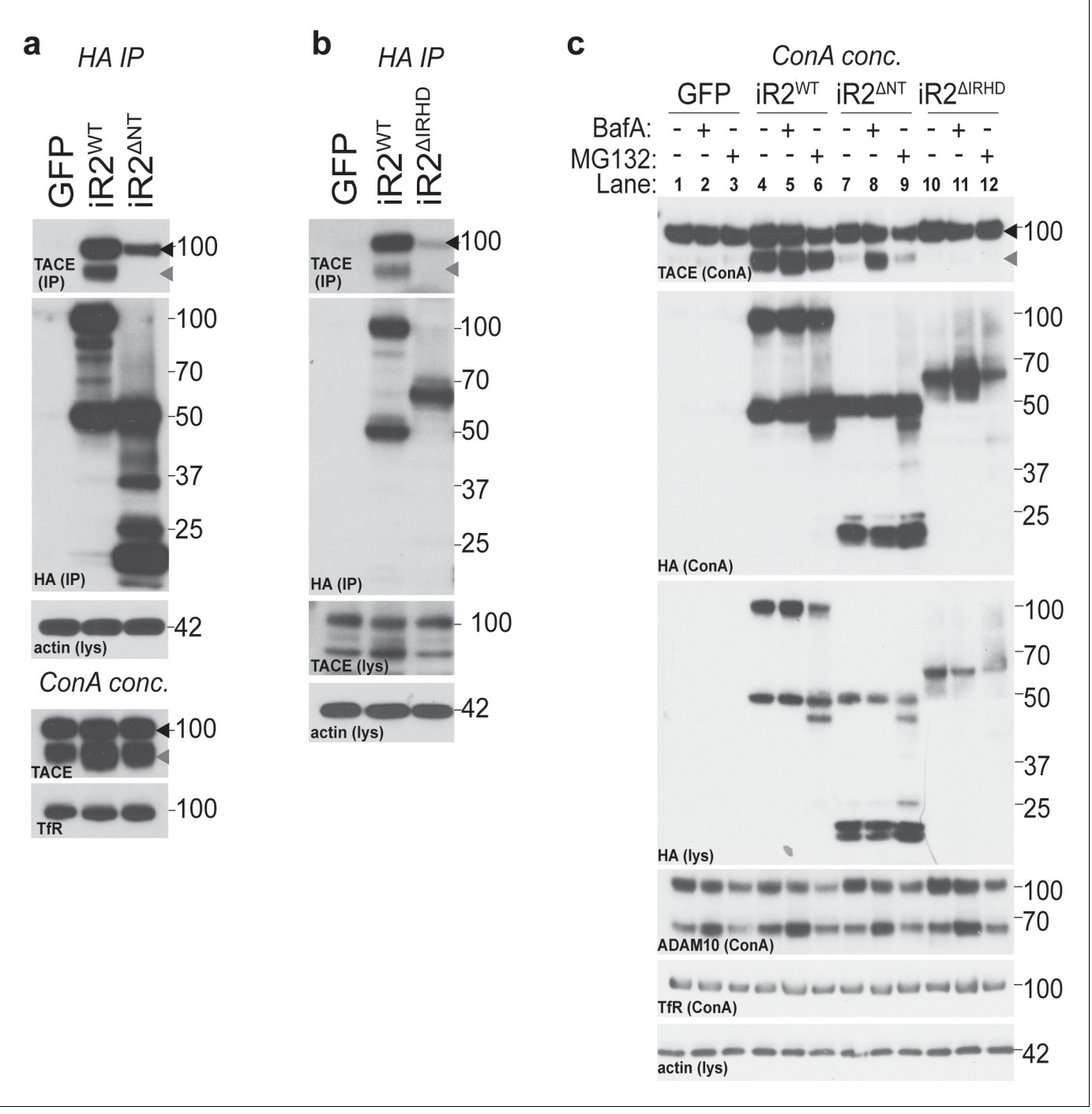

**Figure 4.** The amino-terminus of iRhom2 is required for binding to, and stabilisation of, mature TACE. (a–b) HA immunoprecipitates (IP) and lysates (lys) from WT MEFs (in a) and DKO MEFs (in b) transduced with GFP and HA-tagged iRhom2$^{WT}$, iRhom2$^{\Delta IRHD}$ or iRhom2$^{\Delta NT}$, probed for HA, TACE and beta-actin. Lysates from WT MEFs, enriched for glycoproteins with Concanavalin A (ConA), show equal levels of mature TACE and transferrin receptor (TfR) (in a). (c) Lysates from DKO MEFs expressing GFP or different versions of HA-tagged iRhom2 constructs were enriched for glycoproteins with ConA and were immunoblotted for TACE, HA, ADAM10 and TfR as a labelling control. Lysates were probed with HA and beta-actin antibody. Where indicated, cells had been treated with the lysosomal degradation inhibitor bafilomycin A1 (Baf A, 100 nM) or the proteosomal degradation inhibitor MG132 (10 μM) for 4 hr. Lane numbers have been added for clarity. All panels: black arrowhead: immature TACE; grey arrowhead: mature TACE.

The following figure supplement is available for figure 4:

**Figure supplement 1.** iRhom2 interaction with TACE, and further characterisation of the deletion mutants of iRhom2.

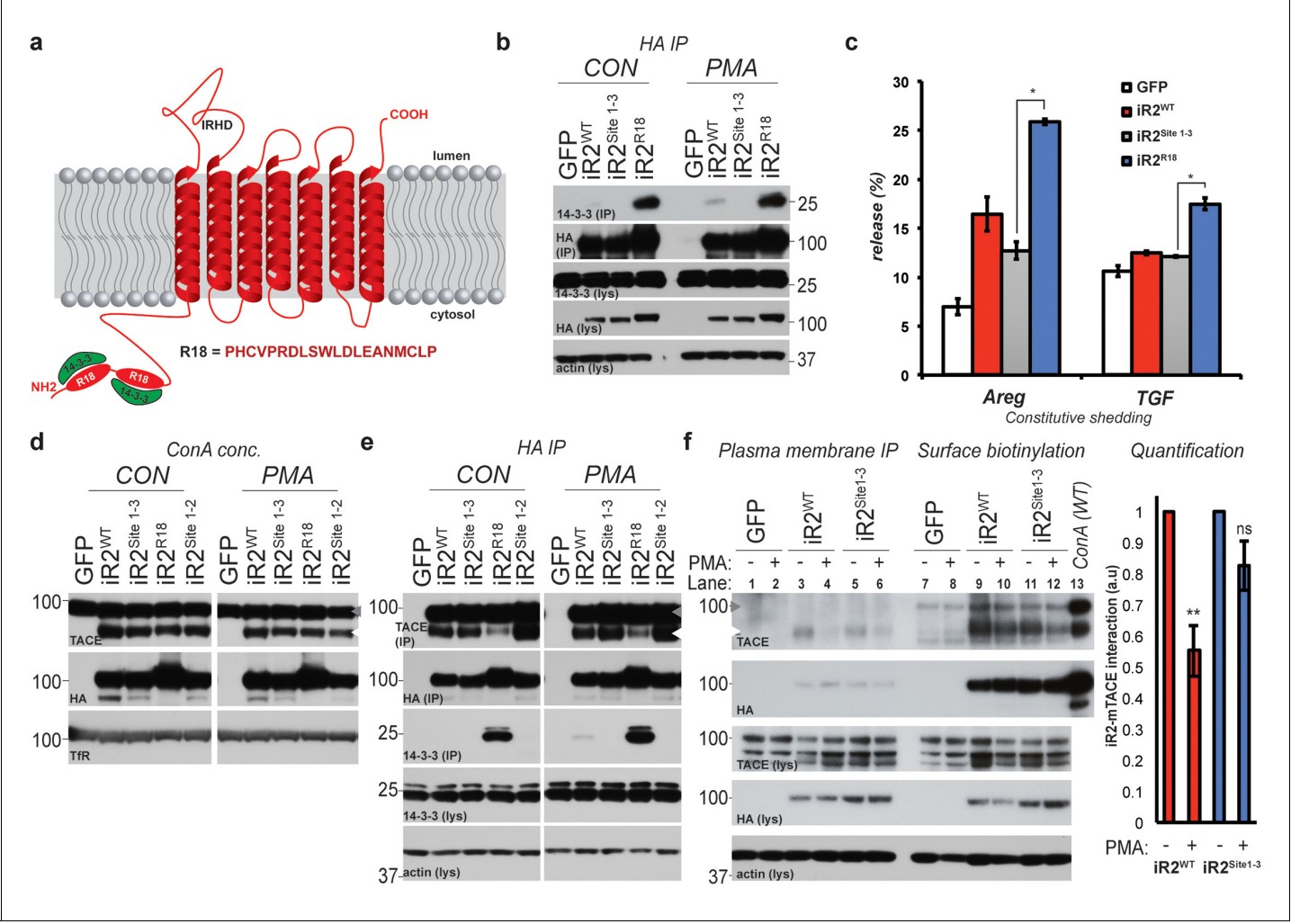

**Figure 5.** Phosphorylation and 14-3-3 protein recruitment uncouples the interaction between iRhom2 and mature TACE to drive its activity at the cell surface. (**a**) A model of mouse iRhom2 (in red), with the two R18 sequence insertions in the N-terminus to facilitate the recruitment of 14-3-3 proteins (in green). Underneath, the sequence of the R18 insertion is indicated. (**b**) HA immunoprecipitates (IP) and lysates (lys) from DKO MEFs transduced with GFP, or HA-tagged iRhom2$^{WT}$, iRhom2$^{Site1-3}$ or iRhom2$^{R18}$, treated with or without 200 nM PMA for 30 min, immunoblotted for HA, 14-3-3 and beta-actin. (**c**) DKO MEFs transduced with GFP or different versions of HA-tagged iRhom2 transfected with alkaline-phosphatase tagged Areg or TGF. Values represent the level of released alkaline phosphatase overnight/total alkaline phosphatase multiplied by 100. Error bars representing standard deviations and values for two-tailed student t-tests, compared to iRhom2$^{Site1-3}$ are depicted. *p<0.05. (**d**) Lysates from DKO MEFs expressing GFP or different versions of HA-tagged iRhom2 enriched for glycoproteins with Concanavalin A (Con A) were immunoblotted for HA, TACE and transferrin receptor (TfR) as a labelling control. Grey arrowhead: immature TACE. White arrowhead: mature TACE. (**e**) HA immunoprecipitates (IP) and lysates (lys) from DKO MEFs transduced with GFP, or HA-tagged iRhom2$^{WT}$, iRhom2$^{Site1-2}$, iRhom2$^{Site1-3}$ or iRhom2$^{R18}$, treated with or without 200 nM PMA for 30 min, immunoblotted for HA, TACE, 14-3-3 and beta-actin. Grey arrowhead: immature TACE. White arrowhead: mature TACE. (**f**) HA immunoprecipitates (left) and neutravidin-enriched cell surface biotinylated protein preparations (right) with corresponding lysates (lys) from DKO MEFs transduced with GFP, or HA-tagged iRhom2$^{WT}$, iRhom2$^{Site1-3}$, treated with or without 200 nM PMA for 5 min, immunoblotted for HA, TACE and beta-actin. As a molecular marker for immature and mature TACE, in the last lane, a lysate from DKO MEFs expressing HA-tagged iRhom2$^{WT}$ enriched for glycoproteins with Concanavalin A (ConA) was loaded. Lane numbers have been added for clarity. On the right-hand side, a quantification of the interaction between iRhom2 and mature TACE from three independent experiments has been plotted. This was achieved by dividing densitometry values of mature TACE by levels of iRhom2$^{WT}$ or iRhom2$^{Site1-3}$, and normalised to the values before addition of PMA (a.u).

The following figure supplement is available for figure 5:

**Figure supplement 1.** Cell surface presentation of iRhom2$^{R18}$.

recognises the extracellular epitope of iRhom2 so that we could specifically isolate plasma membrane iRhom2 before and after (5 min) PMA stimulation. This showed that iRhom2 and mature TACE do indeed form a complex at the plasma membrane, and that this interaction is reduced upon PMA stimulation (*Figure 5f*, compare the reduction in TACE co-immunoprecipitation in lanes 3 and 4, and in lanes 5 and 6; relative to total surface levels of mature TACE in lanes 9–12). Furthermore, the complex is more stable with the non-phosphorylatable mutant, iRhom2$^{Site1-3}$ (*Figure 5f*, see graph for quantification). These data also make the converse point: a more stable iRhom2-TACE complex leads to less TACE activity.

Overall, these results lead us to propose a new signalling mechanism that governs TACE activity. iRhom2 binds to and stabilises mature TACE at the plasma membrane but this complex inhibits TACE activity. Phosphorylation of the iRhom2 cytoplasmic domain drives the recruitment of 14-3-3 proteins, weakening the interaction between iRhom2 and TACE, an essential requirement for TACE proteolytic activity as a shedding enzyme.

## Acute regulation of TACE by iRhom2 phosphorylation controls TNFα release in primary macrophages

TACE and its cytokine and growth factor substrates are profoundly implicated in pathophysiology. One of its best-understood and most biologically important roles is as the primary controller of inflammation through TNFα release (*Black et al., 1997*; *Moss et al., 1997*). iRhom2 is the sole iRhom in macrophages, and iRhom2 KO mice are viable but compromised in macrophage secretion of TNFα, with consequent inflammatory defects (*Adrain et al., 2012*; *McIlwain et al., 2012*; *Siggs et al., 2012*). We therefore investigated whether the phosphorylation-dependent regulatory mechanism we have discovered participates in regulating inflammation in vivo. We reconstituted primary bone marrow-derived macrophages from iRhom2 KO mice with iRhom2$^{WT}$ and iRhom2$^{Site1-3}$ transgenes. TACE maturation was rescued by both constructs, indicating that, as in MEFs, phosphorylation of the cytoplasmic domain of iRhom2 is not needed for TACE trafficking and maturation in vivo (*Figure 6a*). Furthermore, in response to PMA activation or lipopolysaccharide (LPS) treatment (which mimics bacterial infection), we observed both iRhom2 serine phosphorylation, and 14-3-3 binding in primary macrophages (*Figure 6b* and *Figure 6c*). Finally, we found that LPS-triggered shedding of TNFα from iRhom2 KO macrophages was rescued by iRhom2$^{WT}$, but not by its phospho-deficient form, iRhom2$^{Site1-3}$ (*Figure 6d*). We conclude that iRhom2 phosphorylation and 14-3-3 binding mediate regulation of TACE activity in vivo, and therefore represent a pathophysiologically relevant mechanism that controls macrophage-mediated inflammation.

## Discussion

We have discovered that iRhom2 regulates the metalloprotease TACE by multiple, distinct mechanisms throughout the secretory pathway. Because of its ability to trigger both cytokine and growth factor signalling, TACE activity is potentially dangerous, so it is unsurprising that it is itself subject to multiple layers of control. What is more unexpected is that iRhom2, a catalytically inactive relative of rhomboid intramembrane proteases, should be so intimately involved in TACE regulation throughout its lifecycle. We and others have already reported that iRhom2 is responsible for the first step of generating mature TACE, controlling its ER to Golgi trafficking (*Adrain et al., 2012*; *McIlwain et al., 2012*; *Siggs et al., 2012*; *Issuree et al., 2013*); we now show that this was only the first of several stages at which iRhom2 controls TACE. Once TACE has been exported from the ER under the influence of iRhom2 (*Figure 7a*), and its inhibitory pro-domain is removed in the trans-Golgi network, interaction with iRhom2 is essential to stabilise it at the plasma membrane, preventing its lysosomal degradation (*Figure 7b*). Stimulation of TACE proteolytic activity is then also mediated by iRhom2: the cytoplasmic N-terminus of iRhom2 is phosphorylated in response to, for example, GPCR signalling, leading to 14-3-3 binding. Our data support a model in which phosphorylation and 14-3-3 binding elicit a change in the interaction between iRhom2 and mature TACE that licenses TACE activity to shed substrates from the cell surface (*Figure 7c*).

The cytoplasmic domain of iRhom2 maintains the plasma membrane stability of mature TACE. Endocytic clearance of mature TACE from the cell surface has been previously demonstrated after stimulation with PMA (*Doedens and Black, 2000*; *Lorenzen et al., 2016*), although more physiological stimuli do not alter the plasma membrane abundance of the protease (*Lorenzen et al., 2016*),

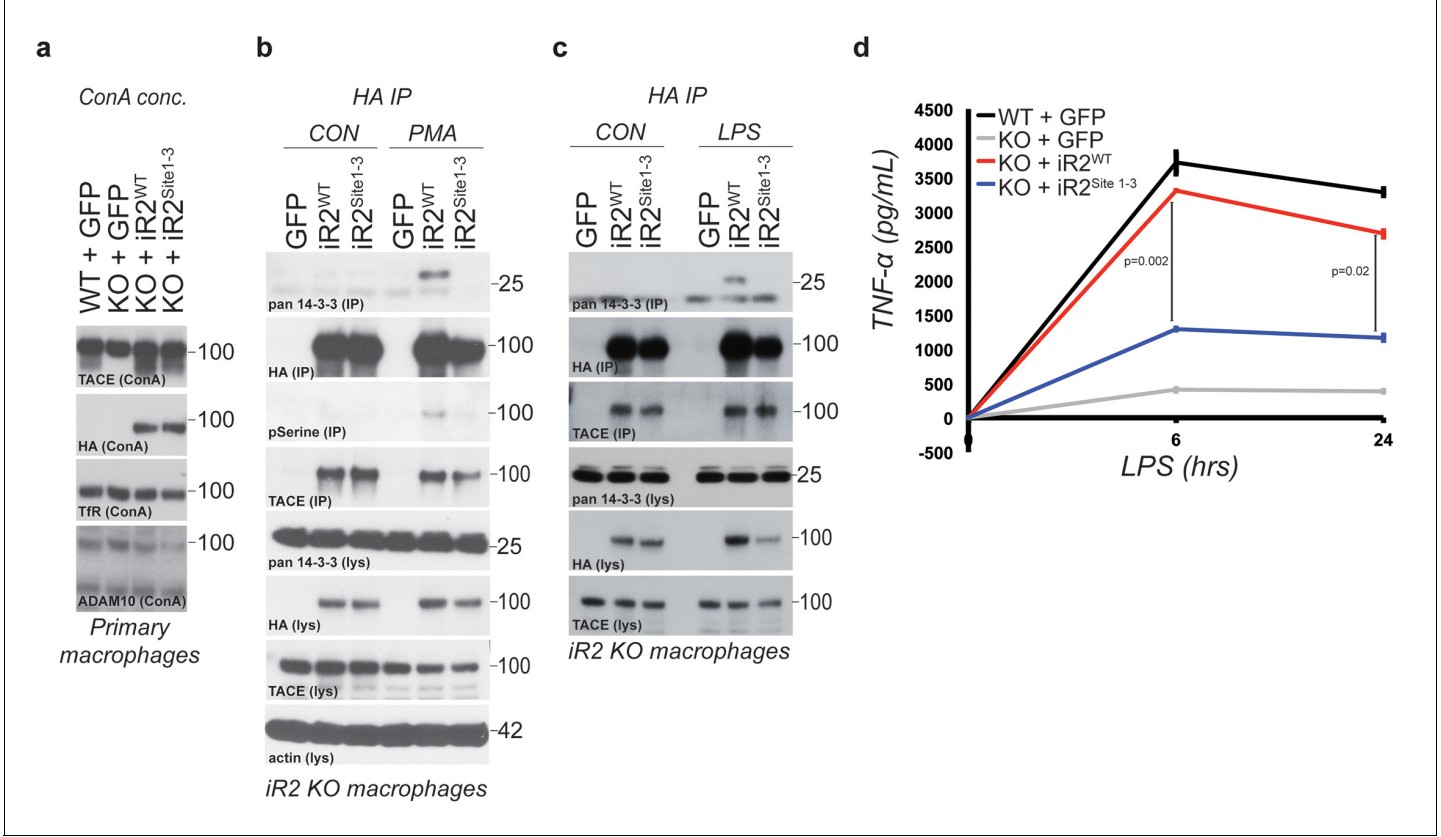

**Figure 6.** Acute regulation of TACE by iRhom2 phosphorylation controls TNFα release in primary macrophages. (**a**) Lysates from WT or iRhom2 KO primary macrophages transduced with GFP, or HA-tagged iRhom2$^{WT}$, iRhom2$^{Site1-3}$, enriched for glycoproteins with Concanavalin A (ConA) and probed for TACE, HA, transferrin receptor (TfR) and ADAM10. NB: The separation of immature and mature TACE is not always apparent in macrophage preparations (*Adrain et al., 2012*). (**b–c**) HA immunoprecipitations (IP) and lysates (lys) from iRhom2 KO primary macrophages transduced with GFP, or HA-tagged mouse iRhom2$^{WT}$or iRhom2$^{Site1-3}$, treated with or without 200 nM PMA (**c**) or 100 ng/ml LPS (**d**) for 15 min, immunoblotted for HA, phosphorylated serine, pan-14-3-3, TACE and beta-actin. (**d**) Levels of secreted TNFα by ELISA from primary macrophages described in **a** treated with 100 ng/ml LPS for indicated times. Error bars representing standard deviations and values for two-tailed student t-tests are depicted.

suggesting that this is not a primary control point of TACE-dependent shedding. Instead, as with many cell surface proteins, there appears to be a constitutive endocytic recycling of mature TACE, mediated by PACS-2 (*Dombernowsky et al., 2015*). The loss of PACS-2 triggers activation-independent mature TACE lysosomal degradation, and it will be interesting to investigate further whether there is a mechanistic link between iRhom2 and PACS-2 in the stabilisation of mature TACE. Interestingly, this resonates with a recent report that iRhom2 maintains the stability of another client protein, the innate immune response adaptor protein, STING (*Luo et al., 2016*), although in this case, STING is degraded by the proteasome.

Signals that lead to the phosphorylation of the cytoplasmic N-terminal domain of iRhom2 cause 14-3-3 binding to iRhom2. The consequence of this signal-dependent 14-3-3 binding is a detectable alteration of the interaction between mature TACE and iRhom2 at the plasma membrane, and activation of TACE-dependent shedding of substrates (*Figure 7c*). Importantly, we have confirmed that this control of stimulated shedding occurs not only in embryonic fibroblasts, but is also an essential regulator of TNFα production in primary macrophages. Furthermore, we find that iRhom2 phosphorylation and 14-3-3 binding is dependent on ERK1/2 signalling, which is itself essential for TACE-dependent TNFα release in macrophages (*Dumitru et al., 2000*; *Eliopoulos et al., 2003*). Consistent with the physiological significance of iRhom2 phosphorylation as a signalling mechanism, Chanthaphavong et al. reported its phosphorylation in primary hepatocytes in response to nitric oxide signalling (*Chanthaphavong et al., 2012*). It is significant that stimulated TACE activity does not

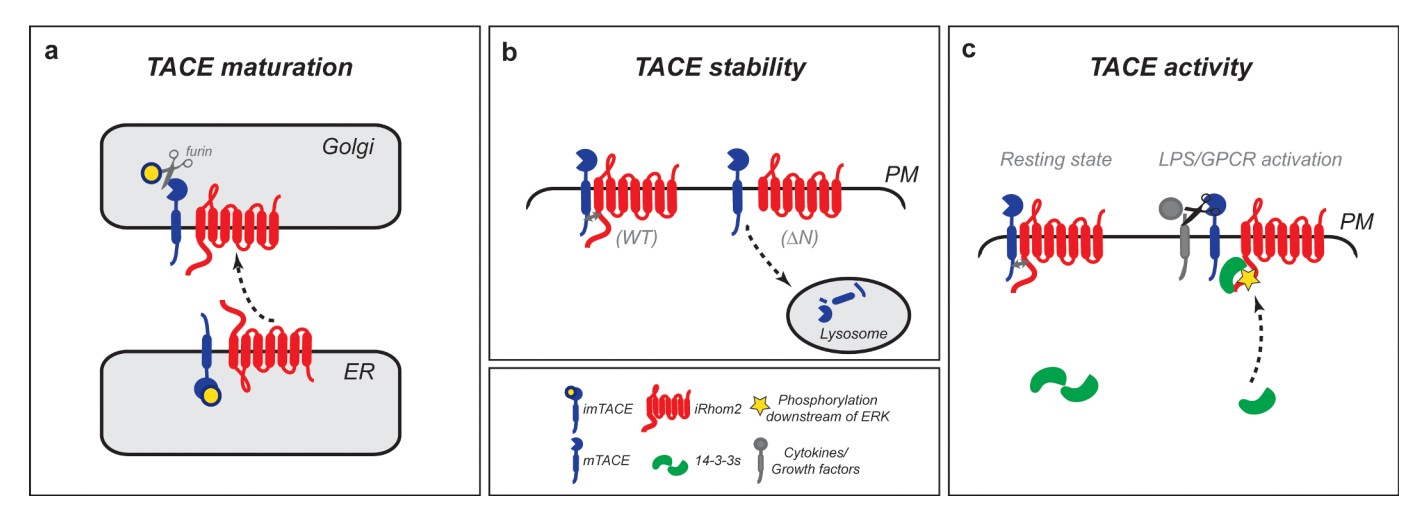

**Figure 7.** iRhom2 regulates TACE at multiple steps in its lifecycle to control growth factor and cytokine signalling. (**a**) iRhom2 and immature TACE must move together from the ER through the Golgi apparatus (**Adrain et al., 2012**) also shown by iRhom2^KDEL (**Figure 1d**), the site of furin cleavage of the TACE prodomain (in yellow), and therefore production of mature TACE. This event requires a contribution from the iRhom2 IRHD. Without iRhom2, there is no mature TACE. (**b**) iRhom2 and mature TACE are in complex with one another at the plasma membrane. The iRhom2 N-terminus stabilises mature TACE levels at the cell surface; in its absence mature TACE is degraded in lysosomes. (**c**) Upon delivery and maintenance of mature TACE at the plasma membrane, GPCR stimulation or macrophage activation with LPS leads to iRhom2 phosphorylation and recruitment of 14-3-3 proteins. The recruitment of 14-3-3 proteins is sufficient to alter the inhibitory interaction between iRhom2 and mature TACE, allowing TACE to cleave its cytokine and growth factor substrates.

depend on its own cytoplasmic domain (**Reddy et al., 2000**; **Horiuchi et al., 2007**; **Le Gall et al., 2010**). Instead, the TACE TMD is required for its rapid activation (**Le Gall et al., 2010**), but the molecular mechanism underlying this observation has been mechanistically obscure. iRhoms have evolved from rhomboid intramembrane proteases, which are polytopic membrane proteins that recognise and cleave transmembrane domains, and it is therefore probable that TMD interaction remains a core function of iRhoms (**Freeman, 2016**). It is possible that the iRhom2-TACE interaction resembles a rhomboid-substrate complex, but that instead of then cleaving the TACE TMD, iRhom2 binding and phosphorylation alters the state of TACE, licensing its activation. Another possible mechanism that reflects the presumed TMD binding role of iRhom2 would be that the phosphorylation of the iRhom2 N-terminus may regulate its binding to TMD-containing TACE substrates, such as TNFα; in other words, that iRhom2 provides a substrate presentation or selection mechanism for TACE.

It will be important to understand how this newly identified phosphorylation-dependent regulatory mechanism relates to other previously identified modes of acute TACE regulation. **Le Gall et al. (2010)** reported that stimulation rapidly alters the accessibility of the TACE active site to a tight binding inhibitor, suggesting conformational changes in TACE itself. It has also been observed that iRhom2 controls substrate selectivity of mature TACE (**Maretzky et al., 2013**). Recently, Sommer et al. provided another model for acute TACE activation (**Sommer et al., 2016**). A variety of external signals cause the translocation of the lipid phosphatidylserine (PS) from the inner to outer leaflet of the plasma membrane. It has been shown that positive amino-acids in the extracellular membrane-proximal domain of TACE interact with the anionic head groups of exposed of PS, and it was proposed that this supports the positioning of TACE to cleave its substrates (**Sommer et al., 2016**). Another model for TACE activation is its conversion from dimers to monomers, which is associated with decreased TIMP3 association (**Xu et al., 2012**). Again, this model has significant overlap with our findings, as it is dependent on p38 MAPK and ERK signalling. The challenge is now to unify these earlier data with our discovery that phosphorylation and 14-3-3 binding of the iRhom2 cytoplasmic domain is necessary and sufficient to trigger TACE-dependent shedding of its substrates. A possible unifying model is that iRhom2 phosphorylation and subsequent conformational change in the

iRhom2/TACE complex is necessary to allow mature TACE to engage with the above events of PS exposure, monomerisation, conformational change of the active site, and substrate selection. It must be stressed that this sequence of events is entirely speculative, but it is consistent with all published data, and does highlight the extraordinarily complex and tight regulation of TACE. Further study of the structure of iRhom2 and TACE should yield further insight into the enzymatic activation process.

Overall, we propose that iRhoms have evolved as long-term molecular partners of TACE: first escorting and controlling TACE secretion and maturation; then stabilising it at the cell surface; and subsequently acting as a signalling hub at the plasma membrane that licenses stimulated TACE activity upon receipt of appropriate signals. These results greatly strengthen the relationship between iRhom2 and TACE, leading us to propose that iRhom2 should perhaps be considered as a multifunctional cofactor or regulatory subunit of TACE.

The incidence of inflammatory diseases is increasing, and there is a need for new anti-inflammatory therapies (*Myasoedova et al., 2010a*, *2010b*). TNFα blocking antibodies are used to treat rheumatoid arthritis, inflammatory bowel disease and psoriasis, and this therapy is one of the biggest-selling pharmaceuticals in the world. It is, however, unsuccessful for approximately a third of patients, requires injection, and is expensive (*Palladino et al., 2003*; *Scott and Kingsley, 2006*; *Esposito and Cuzzocrea, 2009*; *Monaco et al., 2015*). By expanding substantially the scope of the relationship between iRhom2 and TACE, the work we report here strengthens the case for the therapeutic potential of targeting iRhom2. Its previously reported role in regulating TNFα signalling by the ER-to-Golgi trafficking and maturation of TACE makes iRhom2 conceptually a valuable target, but that function will be hard to target pharmaceutically as it depends on a TMD-based interaction in the ER (our unpublished observation). In contrast, the regulatory mechanism between iRhom2 and mature TACE at the plasma membrane described here is cytoplasmic and thus perhaps more targetable. Moreover, the molecular components such as 14-3-3 proteins and the kinases that directly phosphorylate iRhom2 are more conventional, and therefore accessible, pharmaceutical targets. Finally, beyond targeting inflammatory cytokine signalling, pharmaceutical manipulation of the iRhoms-TACE interface could also influence growth factor signalling through the EGF receptor family, which is heavily implicated in a variety of diseases, most notably cancer (*Kenny and Bissell, 2007*; *Ciardiello and Tortora, 2008*).

## Materials and methods

### Reagents
LPS, PMA, Bafilomycin A1, MG132 were all purchased through either Sigma-Aldrich or Roche. U0126 was purchased from Abcam. BI-D1870 was purchased through Cambridge Scientific. Go6973 and Go6983 were purchased from Calbiochem. All drug concentrations are indicated in figure legends and in respective methods sections.

### Antibodies
The following antibodies were used: mouse anti-beta-actin (Santa Cruz, catalogue number sc-47778; WB 1:2000), rabbit anti-TACE/ADAM17 (Abcam, catalogue number ab39161; WB 1:2000), rabbit anti-HA (Santa-Cruz, catalogue number sc-805; IF 1:1000), mouse anti-HA (ENZO, catalogue number ENZ-ABS120-0200; PM IP 1:1000), mouse anti-transferrin receptor (Invitrogen, catalogue number 13–6800; 1:1000), rabbit anti-phosphoserine (Invitrogen, catalogue number 618100; WB 1:1000), rat HA-HRP, clone 3F10 (Roche, catalogue number 12013819001; WB 1:2000), mouse anti-p97 (Pierce/Thermo, catalogue number MA1-21412; WB 1:1000), rabbit anti-pan14-3-3 (Cell Signalling, catalogue number 8312; WB 1:1000), mouse anti-GM130 (BD Transduction labs, catalogue number 610823; IF 1:1000), rabbit anti-pan-cadherin (Cell Signalling, catalogue number 4068S; WB 1:1000), rabbit anti-ADAM10 (Abcam, catalogue number ab1997; WB 1:1000), rabbit anti-ERK1/2 (Cell Signalling, catalogue number 9102, WB 1:1000), rabbit anti-pERK1/2 (Cell Signalling, catalogue number 4377, WB 1:1000). Corresponding species-specific HRP or fluorescently coupled secondary antibodies were used from Santa Cruz and Cell Signaling (WB) or Invitrogen (IF).

## Molecular biology

pM6P.blast-GFP was a kind gift from Felix Randow (LMB, Cambridge). pM6P.blast iRhom2-3xHA and iRhom2$^{cub}$-3xHA has been described previously (Siggs et al., 2014). pM6P.blast iRhom2-3xHA was used as a template for generation of the described individual and combined alanine point mutant constructs e.g. iRhom2$^{Site1}$, iRhom2$^{Site2}$, iRhom2$^{Site3}$, iRhom2$^{Site1-2}$, iRhom2$^{Site1-3}$ and iRhom2$^{pDEAD}$. Primers were designed using the Agilent Technologies website. QuickChange mutagenesis of putative phosphorylation sites was performed according to manufacturer's instructions (Agilent Technologies). PCR products were treated with DpnI at 37°C for 2–3 hr and transformed into XL10 Gold ultracompetent cells (Agilent Technologies). Single colonies were picked and verified by DNA sequencing. For iRhom2$^{R18}$, we synthesised a gBlocks gene fragment (Integrated DNA Technologies) with the insertion of nucleotides coding for amino acids PHCVPRDLSWLDLEANMCLP in the place of regions coding for S58-V59-S60 and S83-L84-S85-Q86-S87. Using this fragment, and an overlapping PCR fragment encoding the remaining sequence of mouse iRhom2, the full iRhom2$^{R18}$ product was generated by PCR. For iRhom2$^{KDEL}$, a reverse primer coding for the KDEL epitope at the 3' end of the 3xHA epitope was used for PCR amplification, using pM6P.blast iRhom2-3xHA as a template. For mouse iRhom2$^{\Delta NT}$, a product that codes for amino acids 367–829 of iRhom2 with sequence coding for 3xHA was PCR amplified using pM6P.blast iRhom2-3xHA as a template. For iRhom2$^{\Delta IRHD}$, using sequential overlapping PCRs, an iRhom2 product lacking nucleotides encoding for amino acids 404–629 was generated. These PCR products were then inserted into digested pM6P.blast constructs by In-Fusion (Clontech) cloning and bacterial transformation, according to manufacturer's instructions. Human unc93B1 and human iRhom2$^{WT}$ or iRhom2$^{\Delta NT}$ (missing nucleotides coding for amino-acids 1–382) plasmids used for large-scale IP and mass spectrometry were amplified from human unc93B1 (BC025669.1; GI:19343917) and iRhom2 cDNA (NM_024599.5; GI:306035188) by PCR and cloned into the NotI cleavage site of lentiviral vector pLEX.puro using Gibson assembly, according to manufacturer's instructions (NEB). All plasmids were verified by DNA sequencing (Source Bioscience, Oxford).

## Cell culture

Mouse embryonic fibroblasts (MEFs) were isolated from $Rhbdf1^{-/-}/Rhbdf2^{-/-}$ (referred to by protein name as iRhom1/iRhom2 KO) E13.5 embryos and wild-type C57BL/6J (RRID: IMSR_JAX:000664) controls, and immortalised by lentiviral transduction with SV40 large T antigen. All MEFs were generated in house and identity confirmed by in-house genotyping. HEK293T cell lines (RRID:CVCL_0063) stably expressing unc93B1 and iRhom2 versions were generated by lentiviral transduction as described in Adrain et al. (2012). HEK293 cells expressing mouse iRhom2-3xHA and mouse iRhom2$^{site1-3}$ were generated by the pLEX-based lentiviral method, exactly as described in Siggs et al. (2014). For mass spectrometry based experiments, HEK293T cells were transduced with pLEX.puro vector containing human unc93B1, human iRhom2$^{WT}$ and human iRhom2$^{\Delta NT}$. Cells were selected by adding 2.5 µg/ml puromycin (Life Technologies). All cells used were maintained in regular high-glucose DMEM, supplemented with 10% FCS, 100 µg/ml penicillin, and 100 µg/ml streptomycin. All cells used in this study were subject to regular mycoplasma testing.

## Bone marrow-derived macrophage isolation and culture

This study was performed in strict accordance with University of Oxford and UK Government rules and guidelines. The procedures and justification for the research was approved under UK PPL 80/2584. Bone marrow was flushed from WT C57BL/6J (RRID: IMSR_JAX:000664) and iRhom2 KO femurs into serum free DMEM/F12 (1:1) (Gibco), passed through 70 µm nylon cell strainers and pelleted at 250 g for 5 min. Cells were resuspended in macrophage growth medium (DMEM/F12 (1:1), containing 10% FCS, 15% of a conditioned culture supernatant from confluent L929 cells, 100 µg/ml penicillin, and 100 µg/ml streptomycin. Cells were grown on non-tissue culture petri dishes, and were maintained and differentiated in macrophage growth medium over 7 days. This resulted in a ~90–95% pure macrophage population as determined by flow cytometry with F4/80 antibody. Prior to experiments, macrophages were detached by PBS/EDTA washes followed by trypsin, and plated at $1 \times 10^5$ in 12 well tissue culture dishes (for ELISA) and at $5 \times 10^6$ in 10 cm tissue culture dishes (for ConA enrichment/HA immunoprecipitation).

## Retroviral transduction of iRhom1;iRhom2 double knock-out (DKO) MEFs and primary macrophages

To generate retrovirus, a pCL-based retrovirus packaging system was used. HEK293 cells grown to 40–50% confluence were transfected with Lipofectamine in 35 mm plates with 1 μg of the pM6P. blast expression plasmids (e.g. pM6P.blast-GFP, pM6P.blast iRhom2-3xHA etc) and 1 μg of the packaging plasmid pCL-Eco. The following day, medium was changed and transfected cells were allowed to secrete virus for 24–48 hr in 2 ml of medium. Culture supernatants were then centrifuged at 20,000 x g, and clarified by filtration with Sartorius Minisart syringe filters (0.45 μm pore size). For infection of target iRhom1;iRhom2 double knock-out (DKO) cells or macrophages, cells were replated the day before, and viral supernatants were diluted 2-fold in fresh medium for transduction. Transduction was carried out in the presence of 50 μg/ml polybrene and a medium change was made 24 hr later. For selection of pM6P plasmids, DKO cells were treated with 5 μg/ml blasticidin 48 hr later. For macrophages, experiments were performed when levels of GFP in pM6P.blast-GFP transduced macrophages had reached detectable levels using a tissue culture fluorescence micro-scope (FLoid Cell Imaging Station; Life Technologies) (on average, 3–4 days post-transduction).

## Immunofluorescence and microscopy

DKO MEFs ($3 \times 10^5$) transduced with indicated constructs were plated on 13 mm glass coverslips in six well dishes. Cells were washed 3x in room temperature PBS and fixed with 4% paraformaldehyde in PBS at room temperature for 20 min. Fixative was quenched with 50 mM $NH_4Cl$ for 5 min. Cells were permeabilised in 0.2% TX-100 in PBS for 30 min and epitopes blocked with 1% fish-skin gelatin (Sigma) in PBS for 1 hr. Coverslips were then incubated overnight with rabbit anti-HA and mouse anti-GM130 in 1% fish-skin gelatin/PBS. After 3x PBS washes, coverslips were incubated with fluores-cently-coupled secondary antibodies (Invitrogen) for 45min. Cells were subsequently washed 3x with PBS and 1x with $H_2O$, prior to mounting on glass slides with mounting medium containing DAPI (ProLong Gold; ThermoFisher Scientific). Images were acquired with a laser scanning confocal micro-scope (Fluoview FV1000; Olympus) with a $60 \times 1.4$ NA oil objective, and processed using Fiji (Image J).

## AP-shedding assay

To test amphiregulin (Areg), transforming growth factor (TGFα), tumour necrosis factor (TNFα), betacellulin (BTC) or epidermal growth factor (EGF) shedding, MEFs were plated at a density of $5 \times 10^4$ in a 24 well plate followed by transfection 16 hr later with alkaline phosphatase (AP) conjugated Areg, TGFα, EGF (kind gifts from Prof. Carl Blobel) or TNFα (a kind gift from Dr Stefan Düsterhöft). For transfection, 200 ng DNA and 0.8 μl of Fugene-6 (Promega) were used, following standard pro-tocols. For GPCR stimulations, MEFs were co-transfected with a construct expressing histamine HA1 receptor (kindly provided by Dr. Stefan Düsterhöft) (400 ng per 24 well). Two days later a stimulation assay was performed as described previously (Christova et al., 2013). In short, cells were washed twice in PBS and incubated in 300 μl Opti-MEM with or without indicated concentrations of PMA (200 nM), or histamine (300 μM) for stated times (Invitrogen). For kinase screening experiments, cells were pretreated with indicated kinase inhibitors for 1 hr in 300 μl Opti-MEM. 100 μl Opti-MEM con-taining 80 nM PMA or 1.2 mM histamine (4x) was then added for 1 hr, in the continued presence of kinase inhibitors. AP activity was detected in stimulated and un-stimulated supernatant or in cell lysates (using Triton-X100 buffer) by adding equal volumes of PNPP buffer (Pierce) and incubating at 37°C followed by measurement of absorbance at 405 nm. The percentage of the total material shed from each well (i.e. signal from supernatant divided by total signal from lysate and supernatant) was then used to calculate the percentage-stimulated release. In the kinase screens, the percentage induction above treatment with 100 μl control Opti-MEM is plotted. The kinase inhibitors Go6973, Go6983, U0126 and BI-D1870 were used at a final concentration of 20 μM. Error bars represent stan-dard deviations, and all two-tailed student t-tests were performed comparing indicated mutants with iRhom2[WT]. In the case of kinase screen experiments, two-tailed student t-tests were performed comparing stimulated AP release in the presence of kinase inhibitors with DMSO treatments.

## ELISA

3–4 days post infection, macrophages were stimulated with 200 nm PMA (Invitrogen) or 100 ng/ml LPS (Sigma) in OPTIMEM supplemented with 1 mg/ml BSA for indicated times. ELISAs were performed with a Quantikine TNF ELISA kit according to the manufacturer's instructions (R&D Systems, catalogue number MTA00B). Error bars represent standard deviations, and all two-tailed student *t*-tests represent a comparison with iRhom2$^{WT}$ data points.

## Immunoprecipitation

For conventional immunoprecipitations (IPs), HEK293 cells or DKO MEFs stably transduced with different versions of iRhom2-3xHA were grown to ~90% confluence in 10 cm plates, on the day of IP. For detection of phosphorylation, or plasma membrane IPs, 24 hr before stimulation, cells were cultured overnight in OPTIMEM. Cells were then stimulated with or without 200 nM PMA in OPTIMEM for indicated times. For plasma membrane immunoprecipitations, after treatment for 5 min, DKO MEFs were placed on ice and washed 3x with ice-cold PBS. Intact cells were then incubated on ice with mouse anti-HA antibody diluted in PBS for 1 hr. Cells were then washed 4x in ice-cold PBS. After all treatments, cells were washed 3x with ice-cold PBS and then lysed in 1 ml TX-100 lysis buffer (1% Triton X-100, 150 mM NaCl, 50 mM Tris-HCl, pH 7.4) supplemented with protease inhibitor cocktail (Roche), 10 mM 1,10-phenanthroline and PhosSTOP (Roche). Cell lysates were cleared by centrifugation at 20,000 x g for 10 min at 4°C. Protein concentrations were measured by a BCA assay kit (Pierce). The lysates were then immunoprecipitated for 2–3 hr with 15 µl pre-washed HA antibody-coupled beads (conventional IPs), or 25 µl ProteinA-coupled agarose beads (Pierce/Thermo) at 4°C on a rotor. After 4–5 washes with lysis buffer, the immunocomplexes were incubated at 65°C for 15 min in 1x LDS sample buffer. Typically, 25% of the immunoprecipitates and 1% of lysates were resolved on SDS-PAGE gels for subsequent western blotting.

## ConA enrichment

As stated in *Siggs et al. (2014)*. For ConA enrichment experiments, cells grown in 10 cm dishes were grown to ~80% confluence were washed twice in ice-cold PBS and then lysed for 10 min in TX-100 lysis buffer (1% Triton X-100, 150 mM NaCl, 50 mM Tris-HCl, pH 7.4) containing complete protease inhibitor cocktail (Roche) and 10 mM 1,10-phenanthroline, to prevent autocatalysis of TACE. Cells/lysates were then scraped and centrifuged at 20,000 x g. Protein concentration was then measured using by a BCA assay (Life Technogies). Lysates were then mixed with washed 50 µl ConA beads (washed in lysis buffer) and incubated for 2–3 hr at 4°C on a rotor. Beads were washed three times in lysis buffer and glycoproteins were eluted with 1 x sample buffer supplemented with 15% sucrose, by incubation at 65°C for 15 min in a thermomixer. 20% of the ConA preparations and 1% of lysates were resolved on SDS-PAGE gels for subsequent Western blotting.

## Cell surface biotinylation

After any indicated treatments, cells were placed on ice and washed 3x with PBS. Surface proteins were labelled on ice with 1 mg/ml Sulfo-NHS-LC-Biotin diluted in PBS (Thermo Scientific, catalogue number 21335) according to manufacturer's instructions. Cells then underwent 1x PBS wash, $1 \times 10$ min incubation in 100 mM glycine/PBS to quench and remove excess biotin, and 3x PBS washes. Cells were lysed in TX-100 lysis buffer (1% Triton X-100, 150 mM NaCl, 50 mM Tris-HCl, pH 7.4) containing complete protease inhibitor cocktail (Roche) and 10 mM 1,10-phenanthroline. After pelleting at 20,000g, clarified supernatants were incubated with 30 µl high-capacity neutravidin agarose beads for 2–3 hr to capture labelled proteins (Thermo Scientific, catalogue number 29204). Beads were then washed 3x with ice-cold lysis buffer and eluted with 1x LDS sample buffer at 65°C for 15 min in a thermomixer.

## Fluorescence activated cell sorting (FACS)

DKO MEFs transduced with HA-tagged iRhom2$^{WT}$ or iRhom2$^{KDEL}$ were detached using 2 mM EDTA and then washed with ice-cold FACS buffer (0.25% BSA and 0.1% sodium azide in PBS). The cells were stained on ice with anti-HA antibody (Santa Cruz rabbit polyclonal (sc-805); 0.5 µg per $5 \times 10^5$ cells) diluted in FACS buffer for 45 min. After two washing steps with ice-cold FACS buffer, the cells were incubated with anti-rabbit Alexa Fluor 488 antibody (Thermo Fisher Scientific donkey polyclonal

(A21206), 1:1000) on ice for 30 min. Cells stained only with the secondary antibody served as control. Cells were washed twice with ice-cold FACS buffer and then analysed with BD FACSCalibur (BD Biosciences) and FlowJo software.

## SDS-PAGE and western blotting

Samples were typically electrophoresed at 150V on 4–12% Bis-Tris gels (Invitrogen) until the dye front had migrated off the gel (approx. 10–15 kDa). Gels were transferred onto PVDF membranes and blocked in PBS or TBS containing Tween 20 (0.05%) and 5% milk or 1% BSA, before detection with the indicated primary antibodies and species-specific HRP-coupled secondary antibodies. Band visualisation was achieved with Enhanced Chemiluminescence (Amersham Biosciences) using X-ray film.

## Mass spectrometry

HEK293T cells were kept on ice and washed with PBS. Proteins were cross-linked in vivo as described in (*Adrain et al., 2012*). Cells were lysed in DDM RIPA lysis buffer (1% n-Dodecyl $\beta$-D-Maltopyranoside (DDM), 150 mM NaCl, Tris-HCl pH 7.5, 50 mM Tris-HCl pH7.5, 0.1% SDS, 0.5% sodium deoxycholate) supplemented with EDTA-free protease inhibitor mix (Roche) and 10 mM 1,10-phenanthroline (Sigma) for 20 min on ice. A centrifugation step (10,000 g, 10 min, 4°C) was performed to precipitate nuclei and insoluble cell fragments. Cell lysates were pre-cleared by incubation with mouse IgG (Sigma) for 1 hr at 4°C. Magnetic anti-HA beads (Pierce) were then added to the supernatant and incubated for 1 hr at 4°C. Beads were washed with DDM RIPA lysis buffer containing 300 mM NaCl, and once with PBS before bound peptides were eluted three times by incubation with HA peptide (1 mg/ml) for 15 min at 37°C. The eluates were pooled and concentrated with a vivaspin concentrator 500 (10,000 kDa MWCO, Sartorius). The spin filter was washed five times with PBS by centrifugation at 12,000 g, RT. Proteins samples were prepared for mass spectrometry using filter-aided sample preparation, as described (*Wiśniewski et al., 2011*). Peptides were eluted from the filter with 0.1% formic acid, 0.1% formic acid in 50% acetonitrile (ACN) and finally with 0.1% formic acid in 80% ACN. The eluate was concentrated and peptides were loaded on PepClean C18 spin columns (Pierce) to remove salt. Peptides were eluted with 0.1% formic acid in 50% ACN and with 0.1% formic acid in 80% ACN and concentrated. Peptides were resolved in 0.1% triflouroacidic acid in 5% ACN and injected into the Ultimate 3000 RSLCnano HPLC (Dionex) – tandem mass spectrometry Q Exactive Orbitrap machine. Data was analysed using the semi-quantitative label-free quantitation method SINQ (*Trudgian et al., 2011*) of the central proteomics facilities pipeline (CPFP) Oxford (*Trudgian et al., 2010*; *Trudgian and Mirzaei, 2012*). Phosphopeptides were identified and analysed with the MASCOT software.

## Acknowledgements

We are indebted to Catherine Rabouille (Hubrecht Institute, NL), Tim Levine (UCL, UK), and Marek Sebesta (Dunn School, Oxford) for critically reading the manuscript; and to Colin Adrain (Gulbenkian Institute, Portugal) for exchange of reagents and ideas. We thank the whole Freeman group for their contributions during the course of this study and feedback on the manuscript. Lewis Taylor (Dunn School, Oxford) helped with FACS confirmation of macrophage populations, and Svenja Hester and Monika Stegmann provided valuable mass spectrometry support. The Freeman group is supported by a Wellcome Trust Senior Investigator Award (grant number 101035/Z/13/Z). AG received funding from the European Union's Horizon 2020 research and innovation programme under the Marie Sklodowska-Curie grant agreement No 659166. HX is supported by the National Natural Science Foundation of China (31640023). UK and BS are supported by Boehringer Ingelheim Fonds PhD fellowships, UK was also funded by the Medical Research Council fund 1374214.

## Additional information

### Competing interests

MF: Reviewing editor, *eLife*. The other authors declare that no competing interests exist.

## Funding

| Funder | Grant reference number | Author |
|---|---|---|
| Wellcome | 101035/Z/13/Z | Adam Graham Grieve<br>Hongmei Xu<br>Ulrike Künzel<br>Paul Bambrough<br>Boris Sieber<br>Matthew Freeman |
| Medical Research Council | Graduate student scholarship | Ulrike Künzel |
| Boehringer Ingelheim Fonds | Graduate student scholarship | Ulrike Künzel<br>Boris Sieber |
| National Natural Science Foundation of China | 31640023 | Hongmei Xu |
| Horizon 2020 Framework Programme | Marie Sklodowska-Curie grant agreement 659166 | Adam Graham Grieve |

The funders had no role in study design, data collection and interpretation, or the decision to submit the work for publication.

## Author contributions

AGG, HX, Conceptualization, Formal analysis, Investigation, Methodology, Writing—original draft, Writing—review and editing; UK, Conceptualization, Investigation, Methodology, Writing—original draft; PB, Conceptualization, Investigation, Methodology; BS, Investigation, Methodology, Writing—original draft; MF, Conceptualization, Supervision, Funding acquisition, Writing—original draft, Writing—review and editing

## Author ORCIDs

Adam Graham Grieve, http://orcid.org/0000-0001-6420-5724
Matthew Freeman, http://orcid.org/0000-0003-0410-5451

## Ethics

Animal experimentation: This study was performed in strict accordance with University of Oxford and UK Government rules and guidelines. The procedures and justification for the research was approved under UK PPL 80/2584.

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
