## [Decision Letter]

Thank you for submitting your article "Phosphorylation of iRhom2 at the plasma membrane controls TACE-dependent inflammatory and growth factor signalling" for consideration by *eLife*. Your article has been reviewed by two peer reviewers, and the evaluation has been overseen by a Reviewing Editor and Vivek Malhotra as the Senior Editor. The reviewers have opted to remain anonymous.

The reviewers have discussed the reviews with one another and the Reviewing Editor has drafted this decision to help you prepare a revised submission.

The manuscript by Grieve et al. investigates how phosphorylation of the rhomboid pseudoprotease iRhom2 controls the activity of the sheddase TACE, which serves as the key activator of TNFalpha and growth factor signaling. The authors add to a flurry of papers published in 2012 revealing an unanticipated role of iRhom2 in regulating ER-to-Golgi-trafficking and subsequent maturation of TACE. In this manuscript, Grieve et al. further elaborate their previous trafficking model, now providing strong evidence that iRhom2 and TACE remain associated throughout the entire secretion route to the plasma membrane. Interestingly, phosphorylation of the N-terminal cytoplasmic domain and subsequent binding of 14-3-3 proteins appears to weaken this interaction thereby increasing TACE shedding activity. Although the molecular mechanism of how TACE activity is controlled in this complex still remains unclear, data presented in this manuscript support an interesting new model that iRhom2 serves as multifunctional subunit of TACE, which simultaneously limits shedding activity and protects TACE from lysosomal degradation. The paper is well written and the experiments are sufficiently explained, but two major conceptual issues and several technical concerns need to be resolved in a revision:

1) An important conceptual gap is the identity of the kinase(s) and how they are controlled. At least the identity of the kinase(s) need to be clarified or narrowed down using well described kinase inhibitors or/and sets of siRNA after physiological (not PMA) stimulation. Without this information the underlying mechanism is not sufficiently developed to warrant publication in *eLife*.

2) The manuscript falls short in defining the role of iRhom2 phosphorylation on complex formation with TACE. From the data presented in the current manuscript (Figure 4), the claim that upon PMA stimulation "the complex is more stable with the non-phosphorylatable mutant" (referred to as Site 1-3 mutant) is not convincingly supported, and serious concerns exist regarding the reliability of this data set. Since x-ray films used for ECL detection of western blots hardly provide a linear readout, the authors need an independent experimental approach to support this biotinylation/co-IP experiment (e.g. proximity ligation assays, or split GFP approach to demonstrate direct physical interaction between TACE and the iRhom2 N-terminal domain). The surface interaction of TACE and iRhom2 should be determined after physiological stimulation (not after PMA treatment). The authors also need to determine whether phosphorylation can also occur in the secretory route or only at the cell surface.

Technical comments:

Introduction:

In order to prevent confusion, the authors need to indicate that TACE also is known as ADAM17.

"[…] with the relatively slow process of ER-to-Golgi trafficking[…]" Define what is meant by relatively slow. Can the authors provide any numbers here? Why do the authors describe this as an "apparent paradox"? Trans-Golgi derived vesicles could also rapidly fuse with the plasma membrane.

The recent paper by Luo et al. describing iRhom2 as an essential factor for STING activity should be mentioned (not only in the Discussion section) suggesting role(s) of iRhom2 next to the regulation of TACE.

Results:

Do the authors also detect other post translational modifications next to phosphorylation? Mono-ubiquitinylation would be very interesting in the context of the suggested role of modulation of iRhom2´s endocytosis.

Subsection “TACE activity, but not TACE maturation, is regulated by iRhom2 phosphorylation”, second paragraph: iRhom´s only function is not only the regulation of ER-to-Golgi transport (see above).

Subsection “TACE activity, but not TACE maturation, is regulated by iRhom2 phosphorylation”, second paragraph: Steady state analysis of mature TACE level (Figure 1 and Figure 5) and cellular localization (Figure 1) do not allow precisely measuring trafficking rates and therefore the conclusion that "any phosphorylation-dependent function must be restricted to an unknown post-TACE maturation role of iRhom2" and "trafficking is phosphorylation independent" need to be toned down.

How do the authors explain the higher molecular weight in the iR2Pdead mutant? It is very surprising that a significant fraction of the ER-retention mutant is also found in GM130 positive compartments similar to the WT and pDead mutants? The ER-retention mutant should also be included in the biotinylation assay shown in Figure 1. How far do these over-expressed iRhom2 variants reflect the physiological intracellular steady-state distribution? Are the authors able to show an effect of the iR2Pdead mutant in comparison with the wildtype protein on an endogenous (not overexpressed alkaline phosphatase tagged) substrate of TACE? Is there any effect of the mutant without induction using PMA? It should be noted (and labelled) that these experiments are performed in mouse fibroblasts using PMA as a non-physiological kinase stimulant.

In Figure 2—figure supplement 1 (and in part also in Figure 2) there is no major differences in TACE activation towards the TGF-α substrate and the authors´ conclusion that this revealed the three distinct sites that contribute to PMA-induced stimulation is unclear. Is the mass spectroscopy analysis performed earlier not a more reliable methodology here?

Subsection “Mature TACE activity is dependent upon iRhom2 phosphorylation at three distinct sites, which co-ordinate binding to 14-3-3 family proteins”, second paragraph: The authors report a proteomic screen, and therefore need to present data showing the entire list of interacting proteins found in this screen. This is an important point.

Do 14-3-3 proteins also bind to iRhom2 upon physiological (non PMA) stimulation?

Figure 3: Only bead and only antibody controls are required for the indicated IP experiments. The multiple bands after HAIP are not explained. Is the (co)transport of the iRhom2 to the cell surface mutants affected? No controls are provided that the used inhibitors (BafA and MG132) worked. Bafilomycin not only affects lysosomal degradation. It would be interesting to analyse the effects of an inhibition of endocytosis or and lysosomal degradation by cathepsins.

Figure 4: improve the labelling of all subpanels.

Subsection “Phosphorylation and 14-3-3 protein recruitment uncouples the interaction between iRhom2 and mature TACE to drive its activity at the cell surface”, first paragraph: Does the R18 peptide sequence inserted in the N-terminus of iRhom2 affect its subcellular trafficking?

Why is the complex of iRhom2 and TACE less stable after expression of iR2Site1-3 mutant and 5 min PMA treatment?

---

## [Author Response]

*[…] The paper is well written and the experiments are sufficiently explained, but two major conceptual issues and several technical concerns need to be resolved in a revision:*

*1) An important conceptual gap is the identity of the kinase(s) and how they are controlled. At least the identity of the kinase(s) need to be clarified or narrowed down using well described kinase inhibitors or/and sets of siRNA after physiological (not PMA) stimulation. Without this information the underlying mechanism is not sufficiently developed to warrant publication in eLife.*

We have addressed this important point extensively, leading to the identification of ERK1/2 as being essential kinases in this process. This has led to the addition of a new Figure 3. Specifically:

Upon re-examination of our mass spectrometry hits for iRhom2 interactors, we found that both ERK1 and ERK2, and RSK3 were significant and reproducible hits. Therefore, we screened their activity with their respective inhibitors.

We found that inhibition of ERK1/2 signalling with U0126, a widely used inhibitor of their upstream kinases MEK1/2, reduced TACE-dependent shedding after stimulation with both the non-physiological stimulant, PMA, and a physiological trigger for TACE activation, histamine.

We analysed 14-3-3 recruitment after TACE activation with PMA and histamine -/+ U0126, and found that inhibition of ERK1/2 phosphorylation strongly and reproducibly inhibited 14-3-3 recruitment to iRhom2.

Analysis of these signalling pathways with U0126 confirmed that both PMA and histamine stimulate ERK1/2 signalling – and that U0126 blocks the phosphorylation of ERK1/2.

Consistent with ERK1/2 functioning upstream of iRhom2, we found that ERK phosphorylation was unaltered in iRhom2 site 1-3 mutants.

We also found that serine phosphorylation of iRhom2 was lost upon treatment with U0126.

These data have led to the creation of a new figure, Figure 3, and an updated Abstract.

In summary, we have found that iRhom2 phosphorylation is dependent on ERK1/2 (though it would take work beyond the scope of this project to prove that iRhom2 is a direct target). We show that PMA and histamine, both of which stimulate iRhom2 phosphorylation-mediated TACE activation, activate ERK1/2. Interestingly, our findings are consistent with previous reports that LPS stimulates ERK1/2, which is required for TNF release in macrophages (Dumitru CD et al., 2000, Cell; Eliopoulos AG et al., 2003, EMBO J). This suggests that ERK1/2 dependent iRhom2 phosphorylation may be a conserved mechanism for TACE activation in macrophages, now mentioned in the discussion.

There is also overlap with another study of TACE activation (Xu P, et al. 2012, Science Signalling), where it was shown that ERK and p38 MAPK signalling lead to TACE activation that coincide with TACE monomerisation and changes in surface presentation. The overlap between this study and ours is covered in the Discussion.

*2) The manuscript falls short in defining the role of iRhom2 phosphorylation on complex formation with TACE. From the data presented in the current manuscript (Figure 4), the claim that upon PMA stimulation "the complex is more stable with the non-phosphorylatable mutant" (referred to as Site 1-3 mutant) is not convincingly supported, and serious concerns exist regarding the reliability of this data set. Since x-ray films used for ECL detection of western blots hardly provide a linear readout, the authors need an independent experimental approach to support this biotinylation/co-IP experiment (e.g. proximity ligation assays, or split GFP approach to demonstrate direct physical interaction between TACE and the iRhom2 N-terminal domain). The surface interaction of TACE and iRhom2 should be determined after physiological stimulation (not after PMA treatment). The authors also need to determine whether phosphorylation can also occur in the secretory route or only at the cell surface.*

Actually, our work does not demonstrate a direct interaction between the iRhom2 N-terminus and the cytoplasmic domain of TACE. Instead, we favour a model in which N-terminal phosphorylation and 14-3-3 recruitment to iRhom2 leads to a change in the TMD-based interaction between iRhom2 and TACE at the cell surface (see Discussion). As iRhom2 and both the pro and mature forms of TACE interact throughout the entire secretory pathway, a selective approach is required to assess the interaction at the cell surface. This is quite difficult, as the cell surface pool is only a small fraction of the total, but it is what we showed in the original Figure 4.

We do accept the reviewers’ point that it would be good to provide a different type of experimental evidence to support the conclusion that there is a change in the interaction upon 14-3-3 binding and we have worked hard to address it.

A split GFP approach will not work because the lifetime interaction between iRhom2 and TACE would mean that the fluorescent GFP signal would be detected at all subcellular sites and is not specific to the small proportion at the plasma membrane.

PLA with unpermeabilised cells, using antibodies that recognise extracellular epitopes of iRhom2 and TACE, is a good idea and should be able to detect only the plasma membrane interaction. Following this suggestion, we have attempted PLA with iRhom2 and TACE, using the extracellular epitope of iRhom2-HA and an antibody that detects the TACE extracellular epitope. However, these experiments were unsuccessful, even in permeabilised cells. We then generated a cell line that expresses iRhom2-HA and TACE-FLAG, with a FLAG epitope placed just before the furin cleavage site- thus allowing us to detect the FLAG epitope at the cell surface. PLA using antibodies against HA and FLAG with this also failed. We therefore conclude that this approach does not work, and suggest that, as in some other cases, steric issues in the complex between iRhom2 and TACE prevent PLA from working – at least with these epitopes.

It is worth restating here that the quantification in the previous Figure 4 (now Figure 5) represent three separate experiments, giving us confidence that the reduction of interaction between iRhom2 and TACE that we detect at the cell surface is robust and reproducible.

Further supporting our interpretation, constitutive 14-3-3 recruitment leads to a loss of interaction between mature TACE and iRhom2. This leads to elevated TACE activity even without PMA treatment. This is fully consistent with the idea that 14-3-3 binding alters the interaction between iRhom2 and mature TACE, thereby stimulating TACE activity.

Nevertheless, we fully accept that without a completely independent experimental approach to this difficult problem, we need to word this conclusion more cautiously, and we have now done this in both the Results and Discussion.

We have additionally addressed the final sentence of this point, about the cellular location of iRhom2 phosphorylation. We performed an experiment with iRhom2-KDEL, to assess whether this mutant of iRhom2, which is restricted to the early secretory pathway and does not support TACE-dependent shedding, could recruit 14-3-3 upon PMA stimulation. We found that iRhom2-KDEL recruited 14-3-3 in a similar fashion to iRhom2-WT (Figure 2—figure supplement 1). We therefore believe that iRhom2 is phosphorylated throughout the secretory pathway, but the only effect of iRhom2 phosphorylation on TACE that we can detect is at the cell surface.

*Technical comments:*

*Introduction:*

*In order to prevent confusion, the authors need to indicate that TACE also is known as ADAM17.*

In the earliest description of TACE in the Introduction, we now state that it is also known as ADAM17 (first paragraph).

*"[…] with the relatively slow process of ER-to-Golgi trafficking[…]" Define what is meant by relatively slow. Can the authors provide any numbers here? Why do the authors describe this as an "apparent paradox"? Trans-Golgi derived vesicles could also rapidly fuse with the plasma membrane.*

Until now, the only known function of iRhom2 in TACE regulation is the ability to promote ER to Golgi transport and maturation of TACE. Although it is formally possible that the TACE activation mechanism could be through gating of trans Golgi derived vesicles, this is shown not to be the case, as phosphorylation-defective mutants of iRhom2 do not affect its own surface presentation, nor the ability of mature TACE to reach the cell surface. The maturation of TACE is already well studied and occurs within a time frame of 3-6 hours (Schlondorff et al., 2000), which is not on the timescale of TACE activation (within 10 mins). We have made a comment to this effect in the manuscript (Introduction, third paragraph).

*The recent paper by Luo et al. describing iRhom2 as an essential factor for STING activity should be mentioned (not only in the Discussion section) suggesting role(s) of iRhom2 next to the regulation of TACE.*

The Luo et al. publication has now been mentioned in the Introduction, in the context of iRhom2 regulation of protein stability (second paragraph).

*Results:*

*Do the authors also detect other post translational modifications next to phosphorylation? Mono-ubiquitinylation would be very interesting in the context of the suggested role of modulation of iRhom2´s endocytosis.*

We do have some preliminary mass spec data potentially detecting mono-ubiquitination, but this has not been confirmed or validated so is not appropriate for publication without further work.

Subsection “TACE activity, but not TACE maturation, is regulated by iRhom2 phosphorylation”, second paragraph: iRhom´s only function is not only the regulation of ER-to-Golgi transport (see above).

Thank you for bringing this error to light. We have correctly reworded this sentence, to make clear the statement is with reference to TACE (subsection “TACE activity, but not TACE maturation, is regulated by iRhom2 phosphorylation”, second paragraph).

*Subsection “TACE activity, but not TACE maturation, is regulated by iRhom2 phosphorylation”, second paragraph: Steady state analysis of mature TACE level (Figure 1 and Figure 5) and cellular localization (Figure 1) do not allow precisely measuring trafficking rates and therefore the conclusion that "any phosphorylation-dependent function must be restricted to an unknown post-TACE maturation role of iRhom2" and "trafficking is phosphorylation independent" need to be toned down.*

We have toned down the first statement to a hypothesis. We have rephrased the final conclusion to make a more specific statement that the ER-to-Golgi traffic of iRhom2 and TACE is independent of phosphorylation, which is supported by our data (TACE still undergoes ER-to-Golgi transport and maturation in iRhom2 phosphorylation mutant cell lines; iRhom2 pDEAD reaches the plasma membrane to a similar level as its wild type counterpart, therefore there is no block in ER-to-Golgi traffic).

*How do the authors explain the higher molecular weight in the iR2Pdead mutant?*

There are a large number of mutations in iRhom2 pDEAD that are likely to affect its overall charge in a manner likely unrelated to phosphorylation. For this reason we performed the screen in the manuscript to create iRhom2 with the site1-3 mutant, which is more minimal that the full pDEAD mutant, and in which this upward shift in molecular weight is not apparent. Depending on the duration of the SDS-PAGE gel that is run, an expected faster migration in the iRhom2 site 1-3 phosphorylation mutants can be observed e.g. HA lysates Figure 3/3F, Figure 5, Figure 6.

*It is very surprising that a significant fraction of the ER-retention mutant is also found in GM130 positive compartments similar to the WT and pDead mutants? The ER-retention mutant should also be included in the biotinylation assay shown in Figure 1.*

The KDEL insertion does not prevent Golgi localisation, it ensures retrieval from the cis-Golgi. We have reworded the text to make this clearer. Both biotinylation experiments and cell surface FACS analysis show a significant reduction in surface levels of iRhom2-KDEL. These data are included in Figure 1—figure supplement 2.

*How far do these over-expressed iRhom2 variants reflect the physiological intracellular steady-state distribution? Are the authors able to show an effect of the iR2Pdead mutant in comparison with the wildtype protein on an endogenous (not overexpressed alkaline phosphatase tagged) substrate of TACE?*

Without an antibody to iRhom2 (despite many attempts), it is difficult to answer the first question. We have avoided gross over-expression to the best of our ability, by ensuring low expression by infection with low titres of virus (aiming for an M.O.I. of 1 virus particle per cell). We provide evidence of an effect of iRhom2 phosphorylation on TACE-dependent release of endogenous TNF in macrophages, by ELISA (Figure 6).

*Is there any effect of the mutant without induction using PMA? It should be noted (and labelled) that these experiments are performed in mouse fibroblasts using PMA as a non-physiological kinase stimulant.*

We had already addressed this important issue in Figure 2 but have now substantially strengthened the evidence that physiological GPCR stimulation of TACE activity is dependent on iRhom2 phosphorylation downstream of ERK1/2 (new Figure 3). This is also demonstrated with LPS-stimulated TACE activity in primary macrophages (Figure 5). In the Results section where we first introduce the use of histamine, we have stated that “PMA is not a physiological trigger for TACE activity”.

*In Figure 2—figure supplement 1 (and in part also in Figure 2) there is no major differences in TACE activation towards the TGF-α substrate and the authors´ conclusion that this revealed the three distinct sites that contribute to PMA-induced stimulation is unclear. Is the mass spectroscopy analysis performed earlier not a more reliable methodology here?*

We demonstrate that sites 1, 2 and 3 independently and significantly affect TACE-dependent shedding in an additive fashion. Within sites 1, 2 and 3, some residues are more critical than others, which is reflected by the lack of major differences in TACE activation in some cases.

*Subsection “Mature TACE activity is dependent upon iRhom2 phosphorylation at three distinct sites, which co-ordinate binding to 14-3-3 family proteins”, second paragraph: The authors report a proteomic screen, and therefore need to present data showing the entire list of interacting proteins found in this screen. This is an important point.*

We now include data from the mass spec screen identifying the kinases involved in iRhom2 phosphorylation (Figure 3). We are reluctant to provide the whole list that was obtained from a different proteomic project, which it is not directly relevant to this project, mainly on the grounds that most of the hits have not been validated. Those that have been are being worked on separately.

*Do 14-3-3 proteins also bind to iRhom2 upon physiological (non PMA) stimulation?*

We include new data demonstrating that 14-3-3 proteins bind to iRhom2 upon stimulation with the physiological stimulus, histamine, in an ERK1/2 signalling dependent manner (Figure 3). This is in addition to the data that were already included, which showed that 14-3-3 proteins are also recruited to iRhom2 upon physiological stimulation of macrophages with LPS, in a phosphorylation-dependent manner (Figure 5).

*Figure 3: Only bead and only antibody controls are required for the indicated IP experiments. The multiple bands after HAIP are not explained. Is the (co)transport of the iRhom2 to the cell surface mutants affected? No controls are provided that the used inhibitors (BafA and MG132) worked. Bafilomycin not only affects lysosomal degradation. It would be interesting to analyse the effects of an inhibition of endocytosis or and lysosomal degradation by cathepsins.*

The interaction between iRhom2 and TACE has already been well documented and shown to be specific. For this reason, we typically use commercially available pre-conjugated HA beads for all immunoprecipitation experiments, that have undergone rigorous in-house testing. Nonetheless, first, we performed a repeat IP with control beads without HA antibody, to demonstrate the specificity of our co-immunoprecipitation of TACE with iRhom2 (Figure 4—figure supplement 1). Second, the beads used in the new Figure 4 are for enrichment of glycoproteins, not immunoprecipitations. All lysates are treated equally and, given the new control provided in Figure 4—figure supplement 1 with beads only, we do not believe there is a need for bead only controls in this ConA experiment (empty beads do not concentrate glycoproteins, such as TACE). Third, we include the whole western blot in these immunoprecipitation experiments so that the respective size of each mutant is clearly demonstrated. The lower molecular weight bands have not been well characterized, but are reproducible and presumably represent degradation products. Fourth, we provide a cell surface biotinylation assay of these mutants to demonstrate their ability to reach the cell surface (Figure 4—figure supplement 1). Fifth, to demonstrate that MG132 has worked, we include a new figure that shows increased ubiquitin levels specifically upon treatment with MG132, demonstrating that the rescue of mature TACE levels is dependent on lysosomal degradation, with no influence from the ubiquitin-proteasome pathway (Figure 4—figure supplement 1). Sixth, in the new Figure 4, we include a panel for ADAM10, which is known to undergo endocytosis and degradation from the cell surface (Marcello E et al., 2013, JCI), showing that bafilomycin treatment leads to an increase in ADAM10 levels. We agree that bafilomycin does not only inhibit endocytic degradation but since it is a specific inhibitor of the vacuolar-type H^+^-ATPase on lysosomes, we prefer to keep to our current wording that mature TACE is degraded in lysosomes.

*Figure 4: improve the labelling of all subpanels.*

We have changed the western blot subpanel labelling from white to black, for clarity.

*Subsection “Phosphorylation and 14-3-3 protein recruitment uncouples the interaction between iRhom2 and mature TACE to drive its activity at the cell surface”, first paragraph: Does the R18 peptide sequence inserted in the N-terminus of iRhom2 affect its subcellular trafficking?*

We now provide a blot from a surface biotinylation assay that demonstrates R18-iRhom2 reaches the plasma membrane in a similar fashion to both iRhom2 WT and Site 1-3, relative to their expression level (Figure 5—figure supplement 1).

*Why is the complex of iRhom2 and TACE less stable after expression of iR2Site1-3 mutant and 5 min PMA treatment?*

Quantification from three independent experiments shows that the iRhom2-mTACE complex is less stable after 5 min PMA treatment; however, there is no statistically significant difference in the stability of the iRhom2-mTACE complex in the iR2 site 1-3 mutant (Figure 5).